# Organizing principles of pulvino-cortical functional coupling in humans

Michael J. Arcaro [1,2], Mark A. Pinsk[1], Janice Chen[3] & Sabine Kastner[1,4]

The pulvinar influences communication between cortical areas. We use fMRI to characterize the functional organization of the human pulvinar and its coupling with cortex. The ventral pulvinar is sensitive to spatial position and moment-to-moment transitions in visual statistics, but also differentiates visual categories such as faces and scenes. The dorsal pulvinar is modulated by spatial attention and is sensitive to the temporal structure of visual input. Cortical areas are functionally coupled with discrete pulvinar regions. The spatial organization of this coupling reflects the functional specializations and anatomical distances between cortical areas. The ventral pulvinar is functionally coupled with occipital-temporal cortices. The dorsal pulvinar is functionally coupled with frontal, parietal, and cingulate cortices, including the attention, default mode, and human-specific tool networks. These differences mirror the principles governing cortical organization of dorsal and ventral cortical visual streams. These results provide a functional framework for how the pulvinar facilitates and regulates cortical processing.

[1] Princeton Neuroscience Institute, Princeton University, Princeton, NJ 08540, USA. [2] Department of Neurobiology, Harvard Medical School, Boston, MA 02115, USA. [3] Department of Psychological & Brain Sciences, Johns Hopkins University, Baltimore, MD 21218, USA. [4] Department of Psychology, Princeton University, Princeton, NJ 08540, USA. Correspondence and requests for materials should be addressed to M.J.A. (email: michael_arcaro@hms.harvard.edu)

The primate cortex comprises a mosaic of functionally specialized regions. A major focus in neuroscience has been to understand how these areas coordinate information to support our perception of the environment. One prominent view is that the thalamus facilitates and regulates communication between cortical areas[1–5]. The thalamus is extensively interconnected with cortex, and, as a general principle, cortical areas that are directly connected to one another are also indirectly connected via the thalamus[2,6]. Thus, anatomically, the thalamus is well positioned to influence cortical function. Within the posterior thalamus, the pulvinar is thought to be critical for active vision by regulating visual cortical processing[6–9]. What are the organizing principles that enable the pulvinar, a relatively small region, to coordinate communication across the large, functionally diverse landscape of visual cortex?

The primate pulvinar is anatomically and functionally heterogeneous. Most of our understanding of the pulvinar comes from studies in non-human primates (Supplementary Note 1)[10], and there is only a beginning understanding in the human brain that the pulvinar appears to be similar to other primate species[11–13]. However, the human brain shows a great number of human-specific adaptations, and it is unclear how they are reflected in the functional organization of the pulvinar, a vastly understudied region of the human brain.

A straightforward set of predictions can be derived from the pulvinar's known anatomical connectivity in non-human primates. The pulvinar's anatomical connections appear to be topographically organized such that neighboring parts of cortex project to neighboring parts of the pulvinar[14]. For example, V1 projections to the pulvinar are in close proximity to V2's projections but at a further distance from V4's projections[6,15,16]. This parallels the observation that neighboring cortical areas tend to be interconnected[17,18], and supports the notion that areas that are directly connected are also indirectly connected via the thalamus[2,6]. Based on such anatomical studies, it might be expected that pulvino-cortical functional interactions reflect cortical distance with neighboring cortical areas interacting with similar parts of the pulvinar. However, many anatomically-distal cortical areas are also interconnected[19,20]. Pulvino-cortical connectivity may reflect functional specialization with cortical regions performing similar computations projecting to the same parts of the pulvinar, regardless of cortical distance. For example, projections from parietal and frontal regions involved in attentional control may overlap in the pulvinar. As such, one might predict that pulvino-cortical functional interactions also reflect the similarity of tuning between cortical regions. Though difficult to discern from anatomy alone, such an organization is predicted based on recent physiological studies in monkeys showing that the pulvinar plays a role in synchronizing activity between cortical regions involved in a visual attention task[9]. Given the pulvinar's heterogeneity, it is possible that these organizing principles (and others) coexist and may vary as a function of subdivision.

While much is known about the anatomical cortical connections in the pulvinar, little is known about the fine-grained functional coupling between the pulvinar and cortical areas. Here, we utilized fMRI in humans and a large variety of experimental tasks to characterize functional organizing principles of the pulvinar. We systematically investigated functional specialization of the human visual pulvinar and its relation to cortical organization. In addition to previously reported retinotopic coding and attentional modulation, we found novel evidence that the human pulvinar is involved in object vision and temporal integration. Clusters selective for visual categories, such as faces and scenes, were identified within the posterior ventral pulvinar, similar to the organization of inferotemporal (IT) cortex. Further, the pulvinar was sensitive to the temporal structure of visual input. Activity in the pulvinar was synchronized across subjects during movie viewing and, within the dorsal pulvinar, the degree of this synchronization was modulated by the temporal continuity of the movie. Functional connectivity analyses revealed the presence of two prominent pulvino-cortical networks. The dorsal pulvinar was functionally linked with frontal, parietal, and cingulate areas involved in attentional control, tool processing, and the default mode network. The ventral pulvinar was functionally linked with occipital and temporal areas involved in form, object, and scene recognition. The organization of cortical coupling within the pulvinar reflected both the distances between cortical areas and their functional specialization. Collectively, these results demonstrate that the dorsal and ventral pulvinar are major nodes in the often cortically-emphasized dorsal and ventral visual processing streams and illustrate that principles guiding the functional organization of cortex are present within the human pulvinar. These data highlight the fine-grained organization of pulvino-cortical interactions and greatly contribute to our understanding of the human thalamus by illustrating important constraints on the role(s) of the pulvinar in facilitating and regulating cortical processing.

## Results

**Regionally-specific tuning in the pulvinar.** Activity within the pulvinar varied as a function of stimulus location, attentional allocation, and context (Fig. 1). The ventral pulvinar responded robustly to visual stimulation (flickering checkerboard) and showed a greater response to contralateral (vs. ipsilateral) visual stimulation while subjects performed a dimming task at a central fixation point (Fig. 1a, left, $p < 0.05$, FDR-corrected). The anatomical extent of these contralateral maps corresponded to the locations of two visual field maps within the ventral pulvinar described previously[11,21]. In contrast, the dorsal pulvinar responded weakly to visual stimulation and did not show a clear visual hemifield preference in individual subjects or in the group average contrast maps. To further quantify the visual responses within the pulvinar, regions of interest within the ventral and dorsal pulvinar were identified based on group average retinotopic maps and foci of functional connectivity with parietal and frontal cortex (as described in the following section; also see[11]) respectively. The degree of contralateral tuning, as assessed by a d prime laterality index contrasting evoked activity from contralateral and ipsilateral visual stimulation (Fig. 1a, right), was greater in the ventral pulvinar as compared with the dorsal pulvinar (t(4) = 5.96, $p = 0.004$, n = 5). Responses in the dorsal and ventral pulvinar also were differentiated based on the degree of attentional modulation. The dorsal pulvinar showed greater attentional modulation than the ventral pulvinar, as assessed by a d prime attentional modulation index contrasting contralateral visual stimulation during covert attention from contralateral stimulation during central fixation (t(4) = 3.16, $p = 0.034$, n = 5). These results are consistent with prior literature showing contralateral tuning and attentional modulation in the pulvinar[22]. Together, these data illustrate a differentiation between the ventral and dorsal pulvinar based on visual responsiveness and attentional modulation, mirroring the broad distinctions of occipital and fronto-parietal cortices.

The pulvinar was also sensitive to the temporal structure of visual stimulation. Sensitivity to temporal structure was assessed by an inter-subject correlation (ISC) approach[23] that measures the temporal similarity of activity evoked by an audiovisual movie across individuals. Responses in both the dorsal and ventral pulvinar during viewing of intact movies were consistent across individuals (Fig. 1b; mean $r > 0.15$, n = 11), demonstrating that

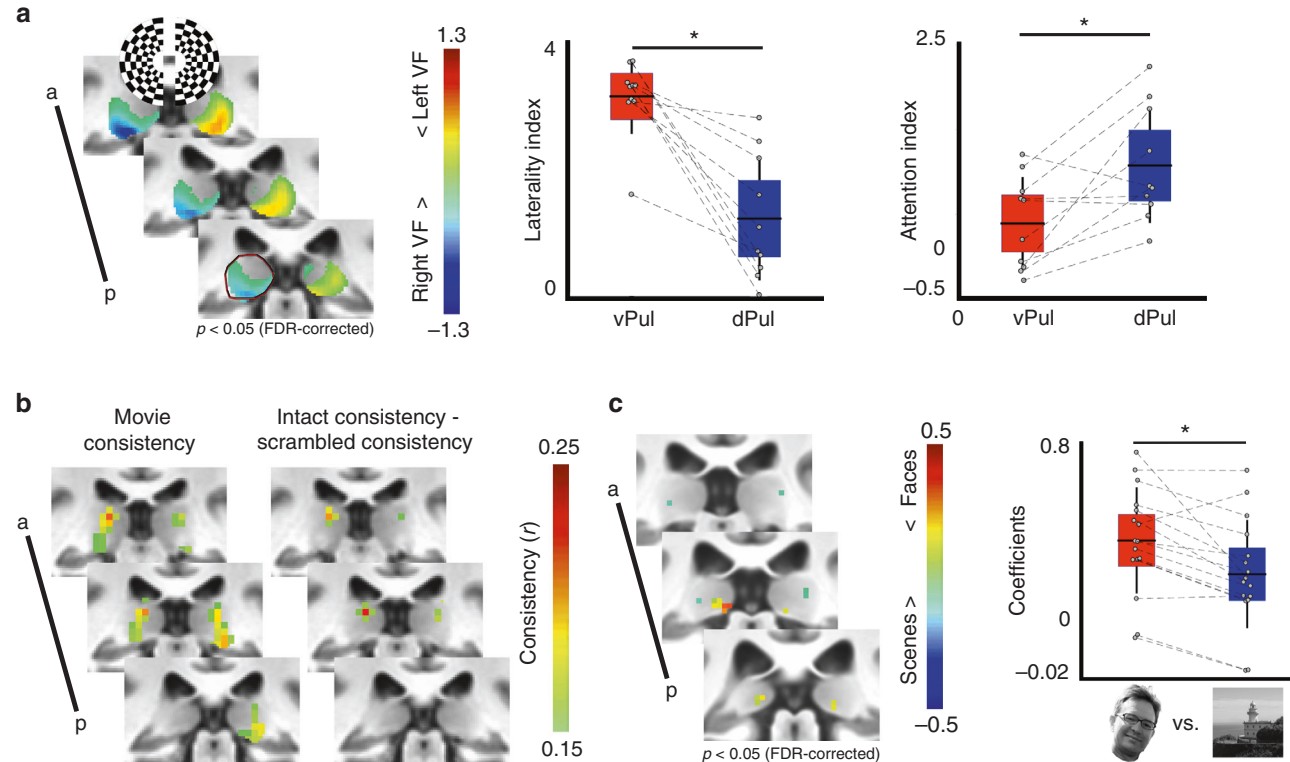

**Fig. 1** Functional distincions within the pulvinar. **a** The ventral half of the pulvinar showed greater responses to stimulation of contralateral visual space (left, $p < 0.05$, FDR-corrected, $n = 5$). Anatomical extent of the left hemisphere pulvinar in posterior-most slice outlined in dark red. Graphs show the group average (horizontal black line), standard deviation (vertical black lines), and 95% confidence interval (shaded area) for laterality and attention indices. Grey circles illustrate individual subjects. The ventral pulvinar (vPul) showed greater laterality of responses (middle, $t(4) = 5.96$, $p = 0.004$, $n = 5$), while the dorsal pulvinar (dPul) showed greater attentional modulation (right, $t(4) = 3.16$, $p = 0.034$, $n = 5$). **b** Repeated presentations of intact movie stimuli evoked consistent responses in both the ventral and dorsal pulvinar. Only the dorsal pulvinar showed greater consistency to repeated presentations of intact vs. temporally-scrambled movies ($r > 0.15$, $n = 11$). **c** A posterior medial portion of the ventral pulvinar responded preferentially to face vs. scene stimuli in the voxel-wise contrast (left, $p < 0.05$, FDR corrected, $n = 16$) and in the ROI analysis (right, $z = 3.10$, $p = 0.002$). Graph conventions same as **a**

the human pulvinar exhibits stereotyped activity during the processing of naturalistic stimuli. However, only the ventral pulvinar showed consistent responses across individuals for viewings of permuted versions of the same movie where temporal structure and narrative flow were disrupted. Subtracting the response consistency of the scrambled movie from the intact movie yielded focal bilateral clusters of voxels within the dorsal pulvinar ($r$ difference $> 0.15$, $n = 11$). These results mirror a previously established hierarchy of temporal receptive windows across occipital, parietal, and frontal cortices[23,24]. That is, activity in the ventral pulvinar reflects moment-to-moment transitions in visual statistics, similar to occipital visual areas, whereas activity in the dorsal pulvinar reflects attentional and contextual information, similar to frontal and parietal areas.

Finally, the pulvinar was sensitive to the category of visual stimulation. Focal bilateral clusters of voxels in the posteromedial, ventral pulvinar showed greater activity for viewing face images than scene images (Fig. 1c, $p < 0.05$, FDR-corrected, $n = 16$). Face selectivity was apparent at the individual subject level. Mean beta coefficients were extracted from this region in individual subjects using a leave-one-out analysis to avoid issues of circularity (Methods: Category localizer ROI analysis). Faces consistently evoked a larger response than scenes in individual subjects (Fig. 1c, $z = 3.10$, $p = 0.002$, Wilcoxon ranked sign test; Cohen's $d = 0.63$). The same region of the ventral pulvinar also showed a preference for faces vs. objects and faces + headless bodies vs. objects + scenes ($p < 0.05$, FDR-corrected, $n = 16$). Across all five categories, faces and headless bodies evoked the largest responses

in the posterior pulvinar as compared with scenes, scrambled images, and objects (Supplementary Figure 1a). The lack of a significant differential between faces and headless bodies may reflect the presence of partially overlapping face- and body-selective regions within the posterior pulvinar that are at a scale finer than our imaging resolution. Indeed, face- (FFA) and body- (FBA) selective regions within fusiform cortex overlap even at high spatial imaging resolutions[25]. Voxels showing a preference for scenes vs. faces were identified lateral to the face/body sensitive pulvinar clusters in both hemispheres. This spatial clustering of visual category information is similar to the organization of ventral temporal cortex where spatially-discrete face- and scene-selective regions are typically identified using the same contrasts (Supplementary Figure 1b) and indicates a role for the posterior ventral pulvinar in high-level visual form processing. The magnitudes of the face vs. scene contrast appeared greater for cortical areas, which might indicate weaker selectivity in the pulvinar. However, it is difficult to make direct comparisons, as (1) the pulvinar is much smaller than ventral temporal cortex, (2) there is an intermixing of neurons and fiber paths in the thalamus (vs. clear distinction of grey and white matter in cortex), and (3) cortex is closer in proximity to the transmit/receive coils. Thus, the apparent weaker selectivity in the pulvinar (vs. cortex) may simply reflect weaker signal sensitivity. Though faces and bodies could be thought of as more 'interesting' than the other stimuli tested, it is unlikely that the effects observed in the pulvinar are driven by arousal or attentive signals. If greater attention had been paid to faces or bodies than to other

stimulus categories then this should have been reflected in areas associated with the attention network (e.g., IPS1-4, FEF, IFS). Across all experiments, the functional response properties of the dorsal and ventral pulvinar mirror functional distinctions between the ventral and dorsal cortical visual streams[26,27].

**Regionally-specific pulvino-cortical functional coupling**. The pulvinar was functionally coupled with visual cortex even in the absence of visual stimulation. We assessed the similarity of within-hemisphere activity between the pulvinar and 39 cortical areas across occipital, temporal, parietal, cingulate and frontal cortices using Pearson correlations. These areas tile visual and associated cortex and were chosen as they share broad functional similarity (i.e., involved in visual processing) yet vary in their specialization and cortical location. Correlations for a subset (23) of these areas were previously reported[11], but analyses were restricted to three subregions of the pulvinar based on retinotopic mapping and network-level analyses evaluating the relationship of functional coupling between cortical areas in the pulvinar were not explored (Methods: *Pulvino-cortical functional connectivity analyses*). For most cortical areas, the mean timeseries of activity was positively correlated with the timeseries of many voxels in the pulvinar. In addition, most cortical areas were positively correlated with each other, making it difficult to evaluate the spatial specificity of the correlated signal. We assumed that signals between directly connected areas should be more correlated than between indirect or weakly connected areas. Therefore, we reasoned that assessing the spatial correlation pattern across all areas should be a more sensitive measure (vs. pairwise temporal correlations) for localizing cortical coupling within the pulvinar. To do this, we first calculated the mean timeseries of activity for each cortical area and computed temporal correlations between all cortical area pairs (referred to as cortical area correlation profile). We then computed the temporal correlation between the mean timeseries of each cortical area with the timeseries of each voxel in the pulvinar (voxel-wise pulvino-cortical temporal correlation profile). For each subject, we then compared the pattern of the voxel-wise pulvino-cortical temporal correlations with the (leaving that subject out) pseudo-group average cortical area correlation profile, yielding a voxel-wise measurement (spatial map) of the similarity between each cortical area's correlation profile and every pulvinar voxel's cortical area correlation profile. This measurement is referred to as the pulvino-cortical connectivity. A complete pulvino-cortical connectivity map within the pulvinar was generated for each cortical area in each subject and was used for all subsequent network-level analyses.

Network-level analyses on the group average pulvino-cortical connectivity ($n = 13$) broadly reflected a functional distinction between dorsal and ventral visual streams (Fig. 2). The similarity between each cortical area's pulvino-cortical connectivity map was calculated, then averaged across hemispheres. Multidimensional scaling (MDS) illustrates the similarity of group average pulvino-cortical connectivity maps between cortical areas (Fig. 2a). Data were clustered using a spectral clustering algorithm[28,29]. The algorithm determined an optimal cluster size of 2. One cluster comprised all occipital and temporal areas (with the exception of a tool-selective area in anterior lateral temporal cortex; labeled latTemp). The other cluster comprised all parietal and frontal regions (with the exception of posterior-most parietal visual map IPS0) as well as the retrosplenial cortex (RSC). For both clusters, most areas were linked with many other areas within the cluster (Fig. 2a) and had weak ($r < 0.15$) or no ($r <= 0$) links between clusters. The notable exception was

RSC, which had several links with both clusters, indicating that it may serve as a bridge between dorsal and ventral clusters.

To visualize if there was any structure to this cluster organization in anatomical space, the difference in cortico-pulvinar functional connectivity between these two cluster maps was calculated for each subject and averaged (Fig. 2b). One-sample two-tailed *t* tests were conducted to identify voxels that showed a consistent difference across subjects ($p < 0.05$ FDR-corrected, Fig. 2b). The group average peak connectivity within the pulvinar for both clusters was symmetrical between hemispheres and extended along the anterior-posterior and lateral-medial axes. The occipital-temporal (blue) cluster was most strongly associated with the ventral pulvinar and the frontal-parietal (red) cluster was most strongly associated with the dorsal pulvinar, mirroring anatomical distinctions previously described in non-human primates[30,31].

The clustering of occipital-temporal and frontal-parietal areas within the ventral and dorsal pulvinar, respectively, held when evaluating functional coupling patterns across the entire cortical surface. Instead of calculating pulvino-cortical connectivity profiles based solely on the similarity between the 39 functionally-localized cortical areas, cortical area correlation profiles were re-calculated for each of these 39 areas based on correlations across a segmentation of the entire cortical surface into 180 areas[32]. Even in this expanded analysis that included many non-visual regions, clustering of occipital-temporal areas and frontal-parietal areas remained (Fig. 2c). Notably, these clusters were less distinct (apparent by the smaller differences in the subtraction maps of Fig. 2c compared to 2b) and contained more cross-cluster links than the clusters from the connectivity profiles restricted to the functionally-defined visual areas. By inspection of the correlation matrices, this appeared to be due to the inclusion of several cortical areas that contained little or no correlation with any of the 39 areas, thereby inflating the similarity of connectivity profiles between the two clusters. This indicates that restricting the analysis to cortical areas that have similar functions to the pulvinar (i.e., areas involved in visual processing) was more sensitive at uncovering the fine-scale structure of pulvino-cortical coupling. Importantly, these broader clusters were still localized to the ventral and dorsal pulvinar, respectively (Fig. 2c), further demonstrating that this is a prominent functional distinction of the human pulvinar.

The structure of pulvino-cortical coupling was consistent across subjects (Fig. 3). The group average pulvino-cortical connectivity profile for each area was similar between hemispheres (mean $r = 0.92 +/-0.02$ across areas) and between averages from subsampling subjects (mean $r = 0.73 +/-0.01$ across areas from 500 iterations of split halves). MDS and clustering on the group average of each hemisphere separately yielded similar structure (Fig. 3a). Further, MDS on each subject's pulvino-cortical connectivity yielded a structure more similar to the pseudo-group average (minus that subject) than random configurations from permutation testing (Fig. 3b; Methods: Multidimensional scaling). The goodness of fit (sum of squared error) from procrustes analysis for each subject (mean sse $= 0.04 +/0.005$ s.e.m. across subjects; max $= 0.08$) was better than goodness of fit measures from permutation testing (99.99% min value across subjects $= 0.62$), indicating that the occipital/temporal and frontal/parietal distinction was consistent across individuals. Further, the goodness of fit for each subject was better than goodness of fit measures from permutation testing where the labels were permuted only within cluster (99.99% min value across subjects $= 0.11$), indicating that the structure within each cluster was also consistent across individuals.

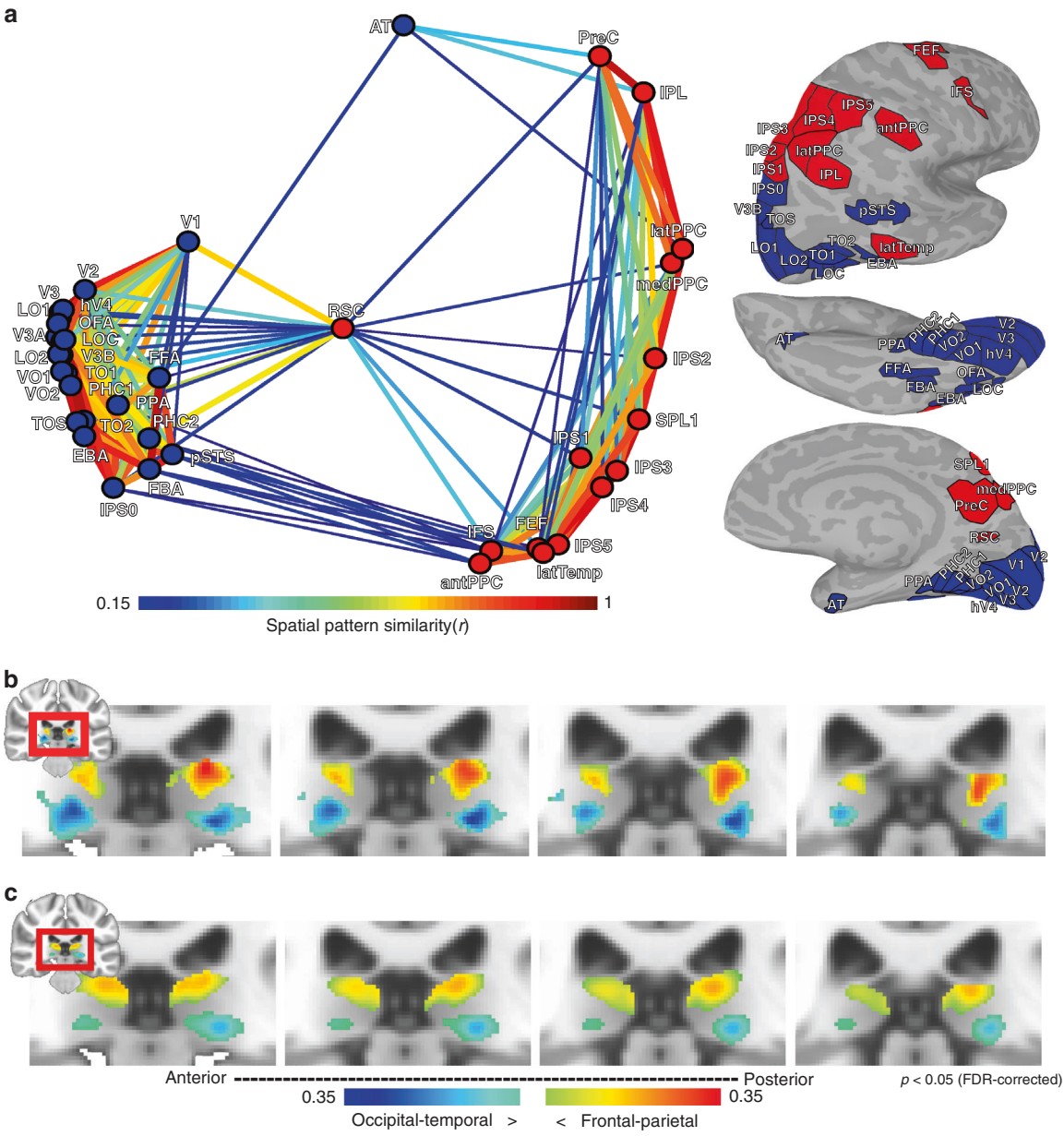

**Fig. 2** Organization of pulvino-cortical functional coupling. **a** 2D plot of multidimensional scaling on the group average ($n = 13$) pattern similarity of pulvino-cortical functional coupling between 39 cortical areas. Only ipsilateral correlations were considered. Data clustering yielded 2 groups. One cluster (blue) mainly contained occipital and temporal cortical areas associated with the ventral visual stream. The other cluster (red) mainly contained frontal and parietal cortical areas associated with the dorsal visual stream. Line thickness and color coding reflects similarity strength between areas. Surface figures illustrate all cortical areas tested; fill color matched to assigned cluster. **b** The difference between the two cluster's pulvino-cortical connectivity maps segmented the pulvinar into dorsal and ventral sections. Colors correspond to the blue and red clusters shown in (**a**), threshold at an p < 0.05 FDR-corrected. **c** Same analysis as in (**b**), but from calculating pulvino-cortical connectivity maps using the Human Connectome Project's 180 cortical areas as the cortical correlation profile

The pulvino-cortical connectivity maps (threshold of $p < 0.05$ FDR-corrected from t-test across subjects) for neighboring cortical areas tended to have a good degree of overlap in the pulvinar and there was a linear relationship between the degree of overlap (Dice's coefficient) and the cortical distance between seed areas (Fig. 4; $r(739) = -0.48$, $p < 0.0001$). This relationship was driven by correlations within the occipito-temporal cluster ($r(274) = -0.63$, $p < 0.0001$) and was not apparent within the fronto-parietal cluster ($r(103) = -0.10$, $p = 0.31$). The difference between the two cluster correlation coefficients was significant ($z = 5.52$, $p < 0.0001$). Taken together, these data demonstrate a broad distinction between the dorsal and ventral pulvinar's

cortical coupling and the functional organization within each subdivision.

**Organization of cortical coupling in the dorsal pulvinar.** Within the dorsal pulvinar, the organization of pulvino-cortical functional coupling reflected the functional topography of parietal, cingulate and frontal cortices. Previously, we showed that parietal area, IPS3, was most strongly correlated with the dorsal pulvinar[11]. Here, we show that individual areas throughout parietal, cingulate, and frontal cortices are most strongly correlated with the dorsal pulvinar, and that these areas are grouped into several subdivisions (Fig. 5a). Several individual

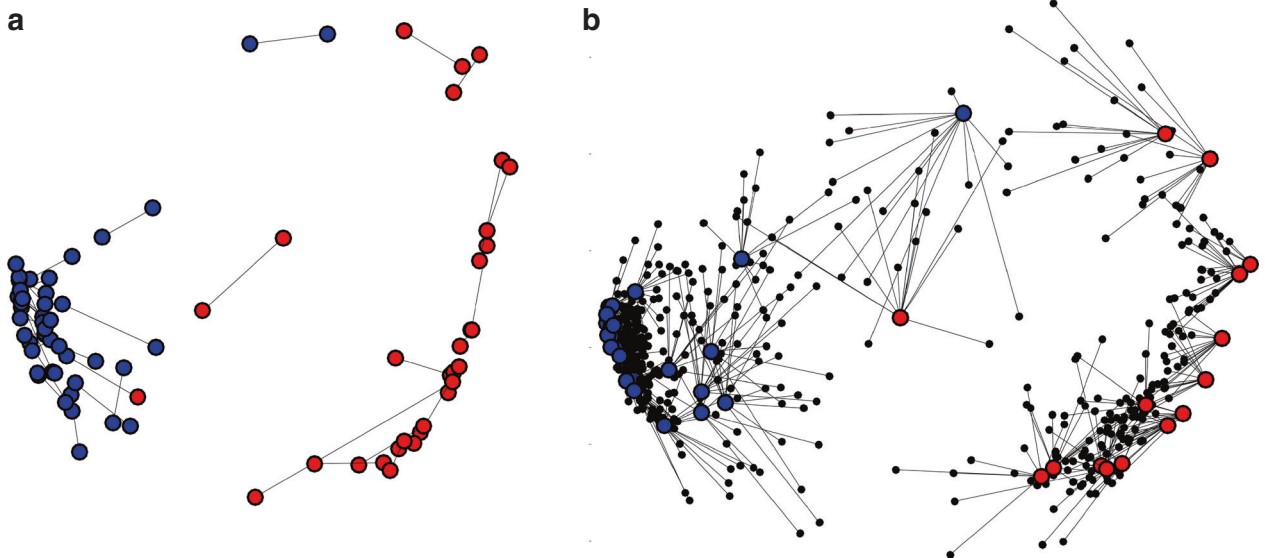

**Fig. 3** Consistency of pulvino-cortical coupling between hemispheres and across individuals. **a** Multidimensional scaling for left and right hemisphere pulvino-cortical functional connectivity. Procrustes analysis was performed to align MDS of the left hemisphere to the right hemisphere. Lines illustrate the distances between areas matched between hemispheres. Dots were color-coded based on clustering performed on each hemisphere separately. The only difference in clustering between hemispheres was area pSTS (i.e., the only blue and red dots linked by a line). **b** MDS for each subject's pulvinar connectivity (averaged across hemisphere). Procrustes analysis was performed to align MDS of each subject to the group average. Small black dots illustrate the locations of individual subject areas. Lines illustrate distances of between individual subjects and the group average for matched areas. Dots were color-coded based on clustering performed on the group average

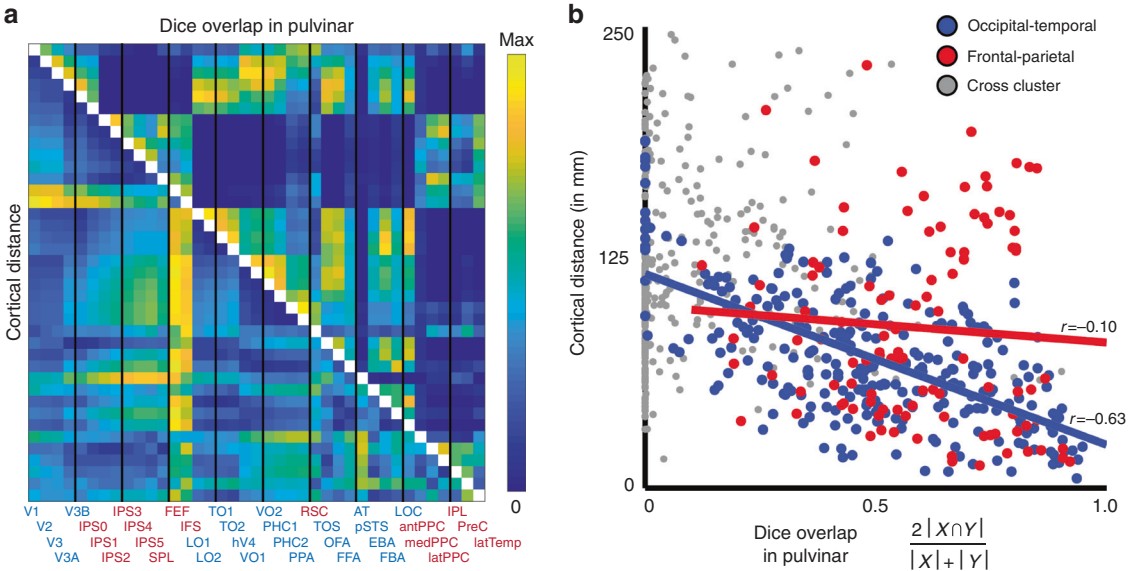

**Fig. 4** Relationship between cortical distance and pulvino-cortical coupling. (left) Half matrices showing pairwise cortical distances and overlap in pulvino-cortical functional connectivity for all 39 cortical areas. Area labels are colored based on clustering in Fig. 2. Thin vertical black lines separate each block of four areas. (right) Scatter plot of cortical distance vs. overlap of pulvino-cortical connectivity for area pairs that were both part of the ventral pulvinar cluster (blue), dorsal pulvinar cluster (red), or for area pairs between clusters (grey). The correlation between cortical distance and pulvino-cortical connectivity overlap was significant for areas within the occipital-temporal cluster ($r = -0.63$, $p < 0.0001$), but not within the frontal-parietal ($r = -0.10$, $p > 0.10$). For all area pairs, Dice's coefficient, $2 |A \cap B| / (|A| + |B|)$, was calculated

pulvino-cortical connectivity maps greatly overlapped even between cortically distant areas such as IPS2 in parietal cortex and FEF in frontal cortex. The foci of these correlations were situated just medial to the lateral edge of the dorsal pulvinar. In contrast, the pulvino-cortical connectivity maps of other dorsal areas appeared to be spatially distinct. For example, the foci of inferior parietal lobule (IPL) and posterior cingulate / precuneus (PreC) correlations fell medial to the IPS and FEF foci and

straddled the medial edge of the pulvinar. Together, these data demonstrate a heterogeneity in the functional organization of the human dorsal pulvinar.

To evaluate the finer-scale structure of these maps, the spatial locations of the peaks in pulvino-cortical functional connectivity were assessed for each cortical area in the dorsal pulvinar cluster. Clustering on the Euclidean distances between the peaks of pulvino-cortical connectivity maps revealed a finer structure of 3

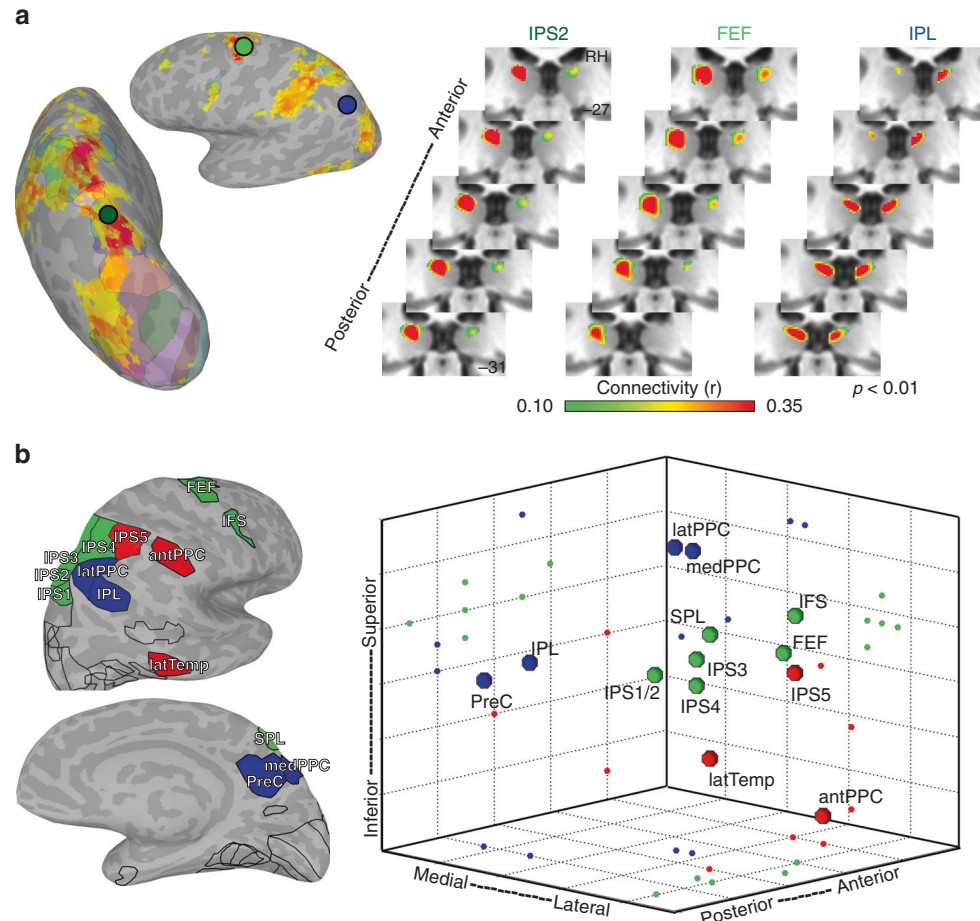

**Fig. 5** Functional coupling with dorsal pulvinar. **a** Group average ($n = 13$; $p < 0.01$ from one-sample $t$ test across subjects) correlations within the dorsal pulvinar for three cortical seed areas (IPS2 (dark green), FEF (light green), and IPL (blue)). Surface figures illustrate the cortical correlation patterns from seed in IPS2 in a single subject. Circle colors correspond to area labels for functional coupling with dorsal pulvinar. **b** Three-dimensional plot of cortical connectivity map peaks in the dorsal pulvinar. Peak coordinates from the left hemisphere were reflected across the midline and averaged with the right hemisphere. Spheres depict the 3D spatial location of each area's peak connectivity within the dorsal pulvinar. 2D projections of each data point are plotted on the walls and floor of the graph. Pulvino-cortical connectivity within the dorsal pulvinar was clustered into three groups. Within the largest cluster (green) containing the IPS1-4, FEF, and IFS, connectivity reflected cortical distance with IPS1/2 located most posteriorly, followed by IPS3/4 and then FEF and IFS located anterior. The peak connectivity for SPL, which is located medial to the IPS maps cortically, was located medial to the IPS1-4 peak correlations in the pulvinar. A second cluster (red) contained tool-selective regions in anterior parietal and lateral temporal cortex as well as IPS5. The third cluster (blue) contained medial, lateral, and inferior parietal areas as well as the precuneus. Surface figures illustrate the outlines of all cortical areas tested and color filled areas correspond to those included in the clustering of the dorsal pulvinar

clusters within the dorsal pulvinar (Fig. 5b). The largest cluster (green) comprised frontal and parietal areas (IPS1-4, SPL, FEF, and IFS) associated with the dorsal attention network (21). Within this cluster, the peaks of pulvino-cortical connectivity maps were topographically organized and reflected the spatial organization of cortical areas. Posterior parietal (IPS1/2) peaks were located most posteriorly in the dorsal pulvinar, followed by anterior parietal (IPS3/4) and then frontal regions (FEF and IFS). The peak for SPL, which is located medial to the IPS maps in cortex, was located medial to the IPS peaks in the pulvinar. The distances between the peaks in this cluster were correlated with the cortical distances between areas ($r(19) = 0.82$, $p < 0.0001$). A second cluster (red) contained IPS5 and regions in anterior parietal (antPPC) and lateral temporal (latTemp) cortex that we, and others, have shown to form a human-specific tool network[33–36]. Within this cluster, there was no clear relation between the distances of pulvino-cortical connectivity peaks and cortical distance. The third cluster (blue) contained two regions in the IPL and the PreC that are associated with the default mode

network[37] as well as two additional regions in medial (medPPC) and lateral parietal cortex (latPPC) (Methods: additional areas). Within this cluster, distances between peaks of pulvino-cortical connectivity reflected functional similarity, not cortical distance. Specifically, the pulvino-cortical connectivity peaks of the functionally similar inferior parietal and posterior cingulate/precuneus areas were within 1 mm of each other, and the other two lateral and medial parietal areas were within 0.5 mm of each other. The peak of latPPC (and medPPC) and the peak of IPL (and PreC) connectivity maps were both separated by several millimeters even though these areas are cortically adjacent. Further, the axes of organization within each cluster did not map across other clusters, suggesting that the organization of areas within a cluster does not translate to other clusters. For example, the anterior-posterior axis reflected cortical distance within the first cluster containing most IPS and frontal maps but did not for the other two clusters. RSC was not included in these analyses since it had several links with both pulvinar clusters (Fig. 2), and therefore was not discretely associated with the dorsal pulvinar.

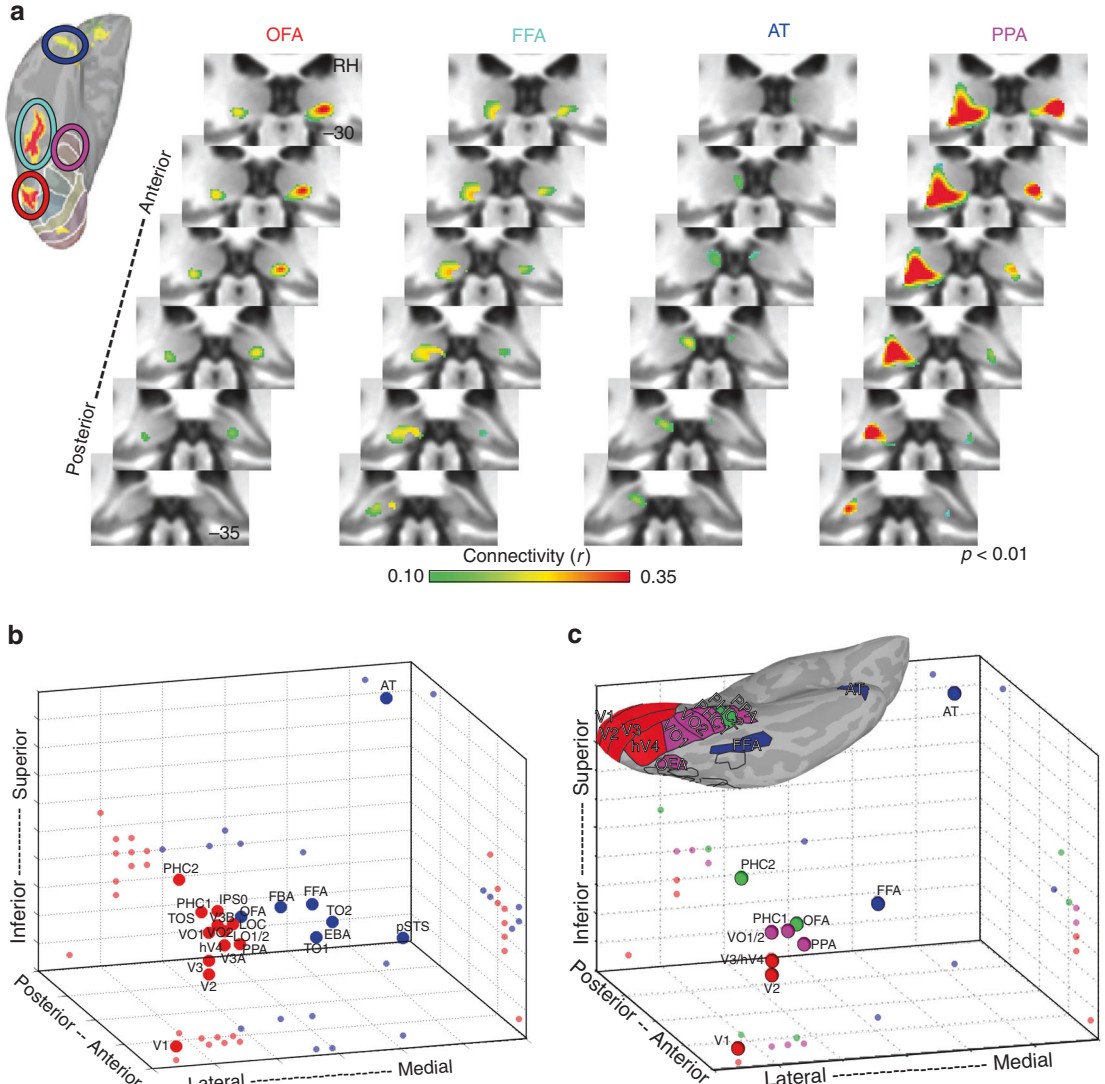

**Fig. 6** Functional coupling with ventral pulvinar. **a** Group average correlation maps ($n = 13$; $p < 0.01$ from one-sample t-test across subjects) are shown for four cortical areas defined based on their functional specialization: occipital face area (orange), fusiform face area (cyan), anterior temporal face area (blue), and parahippocampal place area (magenta). Correlations were strongest within the ventral and posterior-most portions of the pulvinar for each area. Surface figure illustrates the localization of face- and scene-selective regions in an individual subject. Circle color code matches area labels on right. Each cortical region showed greater activity for face vs. scene stimuli ($p < 0.01$, FDR-corrected). **b** Three-dimensional plot of the peak occipital and temporal pulvino-cortical connectivity within the ventral pulvinar. Spheres depict the 3D spatial location of each area's peak connectivity within the ventral pulvinar. 2D projections of each data point are plotted on the walls and floor of the graph. Clustering revealed 2 groups that differentiated lateral (red) and medial (blue) portions of the ventral pulvinar. **c** Same plot as in (**b**), but for a subset of ventral areas. Clustering of connectivity within the ventral pulvinar revealed 4 groups

Last, it is notable that lateral (latPPC) and medial (medPPC) parietal areas, typically not considered to be part of visual cortex, were clustered together and distinct from other visual areas, suggesting that within the dorsal pulvinar there is a distinction in connectivity between visual and non-visual cortical regions. Taken together, these data suggest that multiple functionally dissociable subregions exist within the dorsal pulvinar and that these subregions mirror the functional specialization of parietal, frontal, and cingulate cortices.

**Organization of cortical coupling in the ventral pulvinar.** Within the ventral pulvinar, the organization of pulvino-cortical coupling broadly reflected both functional specialization and cortical distance. Previously, we showed that four occipital-temporal areas, V1, V2, hV4, and TO1, were each correlated

with the ventral pulvinar[11]. Here, we show that two dozen functionally-defined areas throughout occipital and temporal cortices were most strongly correlated with the ventral pulvinar and broadly differentiated lateral from medial portions as well as anterior from posterior portions (Fig. 6). Functional coupling of early visual areas V1-hV4 was localized to anterior parts of the ventral lateral pulvinar, which overlap retinotopic areas vPul1-2[11]. Functional coupling of motion-sensitive cortical areas TO1/2, face-selective area pSTS, and the body-selective area EBA was localized to the ventral medial pulvinar (Supplementary Note 2). Functional coupling of face-selective areas (OFA, FFA, and AT) and scene-selective area PPA was localized to the ventral posterior pulvinar (Fig. 6a). Notably, the FFA's functional coupling was localized to the same postero-medial region of the ventral pulvinar that showed face-selective activity (Fig. 1c) and the spatial patterns

of functional coupling for face-selective areas FFA, AT, and pSTS were each correlated with the spatial pattern of the face-minus-scene coefficient contrast (rs > 0.34) more than any of the other 36 areas (Supplementary Figure 2), demonstrating that pulvino-cortical coupling during rest is predictive of pulvinar activity during visual stimulation. However, overlap between pulvino-cortical connectivity maps was largest for cortical areas in close anatomical proximity regardless of functional specialization. For example, pulvino-cortical connectivity maps of face-selective areas OFA and FFA overlapped with the anatomically-proximal scene-selective area PPA but were separate from the anatomically-distant face-selective area AT. Thus, in contrast to the dorsal pulvinar, pulvino-cortical connectivity maps within the ventral pulvinar also differentiated functionally similar regions.

To evaluate the finer-scale structure of these maps, the spatial locations of the peaks in pulvino-cortical functional connectivity maps were assessed for each cortical area in the ventral pulvinar cluster. There was an anterior-lateral to posterior-medial gradient in the location of pulvino-cortical connectivity peaks from posterior occipital (V1) to temporal (AT) cortical areas. Clustering of pulvino-cortical connectivity within the ventral pulvinar revealed 2 clusters segmenting lateral and medial portions (Fig. 6b). This clustering grouped functionally similar cortical areas. Face-selective areas FFA, OFA, pSTS, and AT were grouped in the medial (blue) cluster and scene-selective areas PPA and TOS were grouped in the lateral (red) cluster. However, the lateral and medial clusters also reflected anatomical distance, differentiating medial occipital-temporal cortex (e.g., V1, VO, and PHC1/2) from lateral temporal cortex (e.g., TO1/2). Notably, the mapping of cortex along the medial-lateral axis was flipped such that medial cortical areas mapped onto the lateral pulvinar and lateral cortical areas mapped onto the medial pulvinar. This flipped mapping between cortical area and pulvinar connectivity appears consistent with broad anatomical connectivity patterns between temporal cortex and the pulvinar in macaques[38]. To further differentiate effects of functional specialization and cortical distance, a subset of areas was selected to directly compare face- and scene-selective ventral regions within ventral temporal cortex as well as early (V1, V2, V3) and intermediate (hV4, VO1/2) visual cortex, which likely comprise the inputs. Clustering on this subset of areas yielded 4 clusters (Fig. 6c). One cluster (red) contained posterior occipital areas V1, V2, V3, and hV4. A second cluster (magenta) contained occipital-temporal areas VO1-2, PHC1, and PPA. A third cluster (blue) contained temporal areas FFA and AT and a fourth cluster (green) contained temporal area PHC2 and occipital-temporal area OFA. Though face-selective areas FFA and AT were clustered together, a third face-selective area, OFA, was clustered with scene-selective area PHC2. The distances between pulvino-cortical connectivity peaks were well explained by the cortical distances across all areas ($r(64) = -0.72$, $p < 0.0001$), suggesting that this was the prominent organizing factor of cortical coupling with the ventral pulvinar. Cortical distance relative to primary visual area V1 best accounted for the distribution of pulvino-cortical connectivity peaks within the ventral pulvinar (Supplementary Note 3; Supplementary Figure 3). Taken together, these data suggest that the spatial layout of cortical functional coupling in the ventral pulvinar appears to be anchored to the first stage of the visual cortical hierarchy (area V1) and may reflect the presence of multiple sub-pathways nested within the broader ventral cortical pathway[39].

## Discussion

The human pulvinar was engaged in a variety of visual tasks. The ventral pulvinar showed strong tuning for contralateral visual space, consistent with previous work demonstrating that the human pulvinar contains at least two retinotopic maps[11,21,40] that are similar to the visual maps of the ventrolateral pulvinar and central lateral region of the inferior pulvinar in other primate species[41–44]. In contrast, the dorsal pulvinar was visually responsive to stimulation of both contralateral and ipsilateral visual space, but showed increased activity during directed spatial attention[45–47] similar to fronto-parietal cortex; also see[46,48,49]. These results are consistent with lesion studies that have shown that damage to the pulvinar affects the allocation of attention, integration of visual information, and filtering of irrelevant information[50–52] as well as with single unit recordings in monkeys that suggest that sub-regions of the pulvinar are involved in spatial attention[53] and reflect awareness rather than solely the physical features of visual stimuli[54]. Our work exceeds previous investigations by identifying functional organizing principles of the human visual thalamus across a large variety of experimental tasks and in relation to cortical networks spanning occipital, temporal, parietal, and frontal lobes. Our report illustrates several novel findings that we will emphasize in the following discussion.

First, the pulvinar is sensitive to temporal structure. Activity in the pulvinar was synchronized across subjects during the viewing of a live-action movie. Interestingly, the temporal window of synchronization differed between ventral and dorsal sub-regions. The ventral pulvinar was synchronized across subjects even when frames of the movie were presented out of order, disrupting the narrative and temporal structure of movie events[23]. This suggests that the ventral pulvinar tracks moment-to-moment variations in low-level visual features. In contrast, the dorsal pulvinar was synchronized across subjects only for intact movie presentations, suggesting that it is generally insensitive to low-level stimulus features. Instead, this indicates that the dorsal pulvinar is involved in cognitive processes such as the tracking of narrative and real-life events that require the integration of information over several seconds and minutes[55]. This divergence in temporal sensitivity between the ventral and dorsal pulvinar parallels the hierarchy of temporal receptive windows spanning early visual cortex to higher-order parietal and temporal regions[23]. Together, these results demonstrate that the thalamus accumulates and processes information at multiple timescales.

Second, the pulvinar encodes high-level visual form information. A posterior medial region of the ventral pulvinar responded preferentially to face (vs. scene) images and was functionally coupled with face-selective temporal areas at rest. A lateral region of the ventral pulvinar showed weak selectivity for scenes (vs. faces) and was functionally coupled with scene-selective temporal areas at rest. These data are consistent with anatomical connectivity studies in macaques that showed projections from the medial ventral pulvinar to the lower lip of the STS[56] and projections from ventral temporal cortex (around area TF) to lateral portions of the pulvinar[38]. Similarly, an electrical stimulation study demonstrated evoked activity within the ventral medial pulvinar from stimulation of face-selective clusters in the lower bank of the STS[57]. Previous studies have demonstrated responsiveness in the monkey pulvinar to object features at the neuronal level[58,59]. Here, we demonstrate that this information is spatially clustered in the human pulvinar similar to the organization of ventral temporal cortex. To our knowledge, this is the first demonstration of such functional clustering within the human thalamus. This is particularly notable given the extensive overlap of cortical projections throughout the pulvinar[6,15,38], which could have had the effect of blurring out such functional specificity. While emphasis is often placed on the pulvinar's role in visual attention, our results illustrate a distinct role in object vision, and more broadly illustrate that ubiquitous principles of

cortical organization (e.g., functional clustering) hold for the thalamus despite considerable differences in architecture (e.g., lack of columnar organization).

Third, the pulvinar was functionally coupled with visual cortex even in the absence of visual stimulation. During rest, the pulvinar was functionally coupled with 39 functionally dissociable areas across occipital, parietal, temporal, frontal, and cingulate cortex. The mean activity of individual cortical areas was correlated with activity in focal regions of the pulvinar. The foci of these correlations tended to overlap in the pulvinar for neighboring cortical areas. Individual areas were functionally coupled to either the dorsal or ventral pulvinar, but generally not to both. The ventral pulvinar was functionally coupled with occipital and temporal cortices, consistent with studies in other primate species showing that these regions are anatomically interconnected[11,14]. The dorsal pulvinar was functionally coupled with parietal, frontal, and cingulate cortices, consistent with anatomical studies in other primates[60–62]. Functional coupling between the pulvinar and dorsal attention network areas (IPS1-4, FEF, IFS) appeared to be situated within lateral parts of the dorsal pulvinar and potentially corresponds to the border between the dorsal lateral nucleus and the lateral extent of the medial pulvinar, similar to other primates[30]. In contrast, the foci of functional coupling with the cingulate and IPL were situated entirely within the medial pulvinar, similar to other primates[63]. This differentiation in cortical coupling, as well as visual tuning, highlights a functional divide between the dorsal and ventral pulvinar similar to what has been observed anatomically in other primates[14,30,31]. More broadly, the differences between the dorsal and ventral pulvinar mirror the divergence of functions between the dorsal and ventral cortical visual pathways[26,27], and illustrate that, similar to cortex, the pulvinar is functionally organized into two major processing streams.

Fourth, the spatial organization of pulvino-cortical coupling reflected cortical distance and functional specialization. Within the ventral pulvinar, cortical correlations predominantly reflected cortical distance. The foci of correlated activity with occipital and temporal areas were distributed along a rostrolateral to caudomedial axis of the ventral pulvinar, similar to the distribution of cortical anatomical connections in the macaque pulvinar[6,15,64]. There was a strong linear relationship between the Euclidean distances of cortical correlation peaks within the pulvinar and cortical distances between seed areas. While some aspects of cortical coupling within the ventral pulvinar were consistent with the functional specializations of occipital and temporal cortex, overall, correlations were better accounted for by anatomical distance. For example, though the foci of correlated activity for two face-selective areas partially overlapped within the ventral pulvinar and were distinct from the foci of correlated activity for a scene-selective area, these data reflected broader differences between ventral-lateral and ventral-medial temporal cortex. The lack of such clustering within the ventral pulvinar solely based on functional specialization may appear at odds with our finding of category selective clusters within the ventral pulvinar, which suggests the presence of functional clustering. However, this likely reflects a difference of spatial scale (local vs. interareal). Local clustering of functionally similar neurons (e.g., a face-selective cluster) reflects a minimization of cortical distance. Therefore, local functional clustering such as face-selectivity within the ventral pulvinar is not at odds with a broader organization of pulvino-cortical coupling based on cortical distance. This is further supported by our finding that the face-selective region in the ventral pulvinar was correlated with a face-selective cortical region (FFA) in temporal cortex. In contrast to the ventral pulvinar, correlations between the dorsal pulvinar and frontal-parietal cortex predominantly reflected functional specialization. Spatial clustering of frontal-parietal correlations within the pulvinar differentiated the dorsal attention network from the default mode and tool networks. Further, the foci of correlated activity for areas such as IPS2 and FEF largely overlapped in the dorsal pulvinar despite their large cortical distances. At a finer scale, cortical distance was apparent in the spatial organization of correlations within the dorsal pulvinar for a subset of frontal-parietal cortical areas. The peaks of correlated activity for dorsal attention network areas were roughly distributed along an anterior-posterior axis within the dorsal pulvinar similar to their relative cortical distances. However, this relationship was not observed for the peaks of correlated activity of other cortical areas within the dorsal pulvinar. Altogether, these results show that the structure of cortical interactions varies throughout the pulvinar and suggest a fundamental difference in how the pulvinar interacts with dorsal and ventral cortical pathways.

The pulvinar incorporates human-specific adaptations. The dorsal medial pulvinar was functionally coupled with anterior parietal (antPPC) and lateral temporal (latTemp) cortical regions involved in the processing of tools. The foci of these correlations were located near the foci of other areas proximal to anterior parietal cortex, but these two areas (along with IPS5) were grouped into a separate cluster, suggesting that the dorsal pulvinar has expanded to allow for interactions with human-specific cortical networks. Despite a substantial increase in size across primates[65], the human pulvinar maintains an organization of nuclei similar to other primate species[65]. Together, these data suggest that the expansion of individual nuclei allows for species-specific functional adaptations within the pulvinar while preserving its global organization across primates.

The pulvinar has been shown to regulate communication within and between cortical visual areas[8,9]. Our results illustrate the fine-grained organization of pulvino-cortical interactions and highlight important constraints on these interactions. We propose that the dorsal pulvinar predominantly facilitates communication between widespread cortical areas that mediate top-down processes such as control of attention, as well as between nodes in the default mode and temporal-parietal tool networks. The ventral pulvinar predominantly facilitates communication between neighboring cortical areas involved in visual recognition and feature extraction, possibly via local competition[66]. Interactions between the dorsal and ventral pulvinar were not prominent in our data with the exception of retrosplenial cortex (RSC), which appeared to be a hub linking the two subdivisions. Additional interactions likely exist and may be mediated by a tertiary area such as the thalamic reticular nucleus (TRN). While not explored in the present study, the monkey pulvinar also contains extensive anatomical connections with other subcortical regions including the superior colliculus[67] and the amygdala[68]. Future work will be needed to resolve how this organization relates to the organization of pulvino-cortical coupling. More broadly, it will be interesting to see whether cortical coupling with the anterior pulvinar and thalamic regions involved in non-visual sensory functions[69,70] exhibit similar principles of organization. Lastly, the relationship between the functional organization of the pulvinar and individual pulvinar nuclei remains to be resolved. Though the localization of functional coupling for individual cortical areas within the pulvinar provides insight into such structure-function relationships, development of high-resolution anatomical MRI techniques that enable the precise delineation of nuclei is needed. Such data would facilitate further comparisons across primate species. Overall, our results demonstrate that the human pulvinar supports a diverse assortment of visual functions, mirroring those of visual cortex, and provide a functional framework for how the pulvinar facilitates and regulates cortical processing.

## Methods

**Participants**. Twenty-eight subjects (aged 20–36 years, seven females) participated in the study, which was approved by the Institutional Review Board of Princeton University. All subjects were in good health with no history of psychiatric or neurological disorders and gave their informed written consent. Subjects had normal or corrected-to-normal visual acuity. Thirteen subjects (S1-S13) participated in task-free resting-state. Sixteen subjects (S2, 3, 8, 12, 14-25) participated in the object localizer experiment. Eleven subjects (S1, 2, 6-13, 26) participated in the movie viewing experiment. Five subjects (S4, 5, 7, 10, 11) participated in the laterality and attentional modulation experiments. To localize subcortical visual field maps, nine subjects (S4, 5, 7, 9-11, 13, 27, 28) participated in a polar angle mapping session and five subjects (S4, 5, 7, 10, 11) participated in an eccentricity mapping session using scanning protocols optimized for subcortical structuresfor more details, see[11]. In addition, all subjects participated in three experiments to localize cortical visual areas: (i) polar angle cortical mapping; (ii) eccentricity cortical mapping; (iii) memory-guided saccade mapping.

**Visual display**. The stimuli were generated on Macintosh G4 and G5 computers (Apple) using MATLAB software (MathWorks) and Psychophysics Toolbox functions[71,72]. Stimuli were projected from either a Christie LX650 liquid crystal display projector (Christie Digital Systems) or a Hyperion PST-100984 digital light processing projector (Psychology Software Tools) onto a translucent screen located at the end of the Siemens 3 T Allegra and Skyra scanner bores, respectively. Subjects viewed the screen at a total path length of ~60 cm through a mirror attached to the head coil. The screen subtended either 30°x 30° (Allegra), or 51°x 30° (Skyra) of visual angle. A trigger pulse from the scanner synchronized the onset of stimulus presentation to the beginning of the image acquisition.

**Resting state**. Thirteen subjects participated in two versions of resting state scans: (1) fixation; and (2) eyes closed[11]. During the fixation scans, subjects were instructed to maintain fixation on a centrally presented dot (0.3° diameter) overlaid on a mean grey luminance screen background for 10 min. During the eyes closed scans, the projector was turned off and subjects were instructed to keep their eyes closed for 10 min. Two runs were collected per resting state condition.

**Movie viewing**. A timescale localizer was used to delineate regions of the pulvinar that were sensitive to short- and long- timescales following established procedures[23,24,73,74]. Eleven subjects viewed an audiovisual movie clip from the 1975 commercial film *Dog Day Afternoon*[75]. Subjects were instructed to attend to the movie and to freely view it. Movie stimuli subtended 20° horizontally and 16° vertically. A 5 min. 45 s clip of the film was presented as well as a temporally scrambled version of the stimulus where the clip was broken into segments spanning 0.5–1.6 s and reordered. Each movie was presented twice.

**Laterality and attentional modulation**. Five subjects participated in an experiment designed to measure contralateral tuning and attentional modulation. Flickering checkerboard stimuli were presented to either the right or left visual hemifield. Stimuli subtended 15° horizontally and vertically. Subjects were instructed either to attend to a central fixation point (0.5°) and to detect changes in its luminance, or to maintain central fixation while covertly attending to one of the hemifields and to detect changes in luminance within a (varying) focal region of the checkerboard. Stimulus conditions were presented in 16 s blocks and were interleaved with 16 s periods where subjects maintained central fixation on a mean grey screen. The subjects' task was varied between blocks. Each stimulus block was presented twice per run. Twelve runs were collected per scan session.

**Retinotopic mapping**. Retinotopic areas V1, V2, V3, V3A-B, hV4, VO1-2, PHC1-2, LO1-2, TO1-2, IPS0-5, SPL1, IFS, and FEF were identified using standard criteria (Supplementary Methods).

**Object localizer**. Face-selective domains (OFA, FFA, AT, pSTS), body-selective domains (EBA, FBA), scene-selective domains (PPA, TOS, RSC), and the object-selective domain LOC were identified using standard criteria (Supplementary Methods). Due to copyright restrictions, the original images used in this localizer cannot be published. In Fig. 1 and Supplementary Figure 1, we present non-copyright images that are representative of the images used in the experiment. The authors affirm that human research participants provided informed consent, for publication of these example images in Fig. 1 and Supplementary Figure 1.

**Additional areas used in pulvino-cortical correlation analyses**. The posterior cingulate/precuneus (PreC) and inferior parietal (IPL) areas were identified based on relative anatomical location and MNI coordinates from prior studies on the default mode network[37]. The lateral temporal (latTemp) and anterior parietal (antPPC) areas were identified based on relative anatomical location and MNI coordinates from our previous study on the tool network[34]. Though these tool-selective regions are often left hemisphere lateralized, we included bilateral ROIs because weak tool-selectivity was observed in homotopic locations within the right hemisphere[34] and these regions are functionally coupled during rest[76]. Lateral

(latPPC) and medial parietal (medPPC) areas were identified as regions located along the cortical surface in-between the IPS2-4 and the IPL and PreC, respectively. MNI coordinates were projected onto each subject's native space anatomical surface. An ROI spanning 6 mm on the surface around each MNI- or anatomically-defined region was drawn for each area. Areas of overlap with surrounding retinotopic maps were excluded. For overlap between latPPC and IPL (as well as between medPPC and PreC), the border was adjusted to the midpoint. ROI size did not greatly affect analyses and results were qualitatively comparable for larger (8 mm) and smaller (4 mm) ROI sizes.

**HCP multi-modal parcellation**. An atlas of 180 areas across the cortical surface of each hemisphere[32] in MNI space was aligned to each subject's native anatomical image using non-linear anatomical registration procedures outlined below. These 180 areas were used in a secondary analysis to evaluate pulvinar functional connectivity patterns based of correlations across the whole cortical surface.

**Data acquisition**. Functional MR images were acquired with a gradient echo, echo planar imaging (EPI) sequence using an interleaved acquisition. The specific parameters for each scan session are outlined below. For the laterality / attentional modulation experiment, data were acquired with a Siemens 3 T Allegra scanner using a circularly polarized head coil (Siemens, Erlangen, Germany). For laterality and attentional modulation experiments, 25 oblique slices were acquired using an EPI sequence with a 128-square matrix (slice thickness 2.5 mm, interleaved acquisition) leading to an in-plane resolution of $1.5 \times 1.5$ mm$^2$ [field of view (FOV) = 256 × 256 mm$^2$; repetition time (TR) = 2.0 s; echo time (TE) = 40 ms; flip angle (FA) = 90°]. For all other experiments, data were acquired with a Siemens 3 T Skyra scanner using 20-channel phased-array head(16)/neck(4) coil (Siemens, Erlangen, Germany). All functional acquisitions used a gradient echo, echo planar sequence with a 64-square matrix (slice thickness of 4 mm, interleaved acquisition) leading to an in-plane resolution of $3 \times 3$ mm$^2$ [FOV = 192 × 192 mm$^2$; GRAPPA iPAT = 2; 32 slices per volume for resting state and 27 for movie stimuli; TR = 1.8 s for resting state and 1.5 s for movie scans; TE = 30 ms; FA = 72°]. High-resolution structural scans were acquired in each scan session for registration to surface anatomical images (MPRAGE sequence; 256-square matrix; 240 × 240 mm$^2$ FOV; TR, 1.9 s; TE 2.1 ms; flip angle 9°, $0.9375 \times 0.9375 \times 0.9375$ mm$^3$ resolution).

**Preprocessing**. The data were analyzed using AFNI Analysis of Functional Neu-roImages, RRID:nif-0000-00259;[77], SUMA[78], FSL FSL, RRID:birnlex_2067;[79] http://fsl.fmrib.ox.ac.uk/fsl/fslwiki/[80,81], FreeSurfer FreeSurfer, RRID:nif-0000-00304;[82,83] http://surfer.nmr.mgh.harvard.edu/, and MATLAB (MATLAB, RRID: nlx_153890). Functional data were slice-time and motion corrected. Motion distance (estimated by AFNI's 3dvolreg) did not exceed 1.0 mm (relative to starting head position) in any of the 6 motion parameter estimates (3 translation and 3 rotation) during any run for any subject.

**Laterality and attentional modulation experiment**. Data were spatially filtered with a 4 mm (FWHM) Gaussian kernel, which increased signal-to-noise ratio (SNR) while maintaining good anatomical localization of signals within the pulvinar (Fig. 1). A multiple regression analysis (AFNI's 3dDeconvolve) in the framework of a general linear model was performed. Each stimulus condition was modeled with square-wave functions matching the time course of the experimental design convolved with a hemodynamic response function. Additional regressors that accounted for variance due to baseline shifts between time series, linear drifts, and head motion parameter estimates were also included in the model. Brain regions that responded more strongly to right or left visual fields were identified by contrasting blocks of checkerboard stimulation to the right or left visual field while the subject maintained a central fixation. Brain regions that responded more strongly during covert attention were identified by contrasting blocks of covert attention to the checkerboard stimuli vs. blocks of central fixation during visual stimulation. In each hemisphere, covert attention contrasts were only considered for attention to the contralateral visual field. Contralateral tuning was assessed by computing a d prime index ($d'$), defined by the following formula (1).

$$d' = \frac{\mu\text{contralateral} - \mu\text{ipsilateral}}{\sqrt{\frac{\sigma^2\text{contralateral} + \sigma^2\text{ipsilateral}}{2}}}, \qquad (1)$$

where $\mu$contralateral and $\mu$ipsilateral are the average responses to contralateral visual stimuli (during central fixation); $\sigma$contralateral and $\sigma$ipsilateral are the SDs. Attentional modulation was calculated using the same formula between covert hemifield attention and central fixation conditions.

**Category localizer ROI analysis**. A leave-one-out analysis was performed to evaluate the profile of responses across visual categories within the pulvinar. For each subject, an n-1 pseudo-group average face-minus-scene contrast map was computed using a mixed effects meta-analysis (AFNI's 3dMEMA) to identify a face-selective region of interest within the posterior medial pulvinar. The mean betas for each category were calculated within this region from the held-out subject's data.

**Additional preprocessing for correlation analyses**. Several additional steps were performed on the data: (1) removal of signal deviation > 2.5 SDs from the mean (AFNI's 3dDespike); (2) temporal filtering retaining frequencies in the 0.01-0.1 Hz band; (3) linear and quadratic detrending; and (4) removal by linear regression of several sources of variance: (i) the six motion parameter estimates (3 translation and 3 rotation) and their temporal derivatives, (ii) the signal from a ventricular region, and (iii) the signal from a white matter region. To avoid partial volume effects with surrounding grey matter, ventricular and white matter regions were identified by hand on each subject's mean EPI image. These are standard pre-processing steps for resting-state correlation analyses[84,85], though our results were not dependent on these preprocessing steps, and correlation analyses on the raw data yielded qualitatively and statistically similar results. Global mean signal (GMS) removal was not included in the analysis reported here given concerns about negative correlations[86–88], though inclusion of GMS removal yielded statistically similar results. To minimize the effect of any evoked response due to the scanner onset, the initial 21.6 s were removed from each rest scan. To extract the mean signal from each cortical area, voxels that fell between the gray and white matter boundaries were mapped to surface model units (nodes). The mean signal from each cortical area was extracted into MATLAB for correlation analyses. Data within the thalamus was then spatially filtered with a 4 mm (FWHM) Gaussian kernel, which increased SNR and improved correspondence of correlation patterns between subjects while maintaining good anatomical localization of signals within the pulvinar.

**Inter-subject correlations on movie data**. For the movie viewing experiment, an inter-subject correlation (ISC) analysis approach was used[23,73]. This analysis provides a measure of the consistency of a response to the temporally complex stimulus (i.e., naturalistic audiovisual movie) by comparing the BOLD response across different subjects. ISC helps avoid the contribution of idiosyncratic responses to correlation patterns and circumvents the need to specify a model for the neuronal processes in any given brain region during movie watching. For ISC, repetitions for each condition were averaged within subject. Data were then transformed to MNI space and voxel-wise correlations were computed between the responses in each subject with the average of all other subjects for each condition separately. This yielded a voxel-wise measure of the consistency of brain activity evoked by the audiovisual movie clip. Group average correlation maps were calculated, and the difference was computed between intact and scrambled correlation maps.

**Pulvino-cortical functional coupling analyses**. We applied a two-step Pearson correlation analysis to identify cortical areas whose spatial profile of correlations with all other cortical areas (referred to as cortical area correlation profile) was similar to the cortical area correlation profile of individual voxels in the pulvinar. First, temporal correlations were performed between each cortical area and between cortical areas and the pulvinar. To identify the cortical correlation profile of individual cortical areas, the mean timeseries of each cortical area was correlated with the mean timeseries of every other cortical area in each subject. To identify the cortical correlation profile of the pulvinar in each subject, the mean timeseries of each cortical area was correlated with the timeseries of each voxel within a pulvinar mask.

Next, we compared the pattern of cortical correlations (profiles) for individual pulvinar voxels with those of individual cortical areas. For each subject, the cortical correlation profile of each pulvinar voxel was correlated to the pseudo-group average cortical correlation profile of each cortical area, where the average excluded that subject. This yielded a measurement of similarity between each cortical area's cortical correlation profile and every pulvinar voxel's cortical correlation profile, which we refer to as the pulvino-cortical connectivity. Significant positive similarity in the pulvino-cortical functional connectivity was taken as evidence for connectivity between a pulvinar voxel and a given cortical area. Individual subject anatomical volumes were then aligned using a two-step linear (AFNI's 3dAllineate), nonlinear (AFNI's 3dQwarp) registration procedure, and each subject's pulvino-cortical functional connectivity profile was transformed into MNI space. Voxel-wise, one sample t-tests were used to assess statistical significance. Even for the 23 cortical areas that were investigated previously[11], the computation of each pulvino-cortical connectivity map differs in the current study due to the inclusion of several (16) additional parietal, temporal, and frontal areas, which changes the size of the cortical connectivity profile. Though the cortical connectivity profile has been greatly expanded in this study, the resulting pulvino-cortical connectivity maps for the 5 individual cortical areas previously reported (V1, V2, hV4, TO1, and IPS3) were very similar to maps from the previous analyses, demonstrating the robustness of these connectivity measures.

**Group average pulvino-cortical connectivity profile**. While functional connectivity profiles for individual subjects are limited in spatial precision by the imaging acquisition resolution, group average maps can yield a finer spatial precision than the original sampling resolution as has been previously demonstrated in group average subcortical imaging[11,89]. The correspondence between the EPI acquisition grid and anatomy naturally varies across individuals; thereby leading to variability in the degree to which individual voxels sample functionally distinct regions in the pulvinar across individuals. Averaging voxels across subjects reduces effects in voxels where the signals are less prevalent across subjects, thereby increasing the spatial precision. The degree to which this averaging improves spatial precision is limited by the accuracy of anatomical registration across individuals. However, nonlinear registration algorithms can handle the alignment of subcortical structures across individuals given the uniformity of subcortical structures (especially in comparison to the variance of cortical folds across individuals). e.g., previously, we showed that pulvino-cortical connectivity maps with V1 overlapped with two prominent retinotopic maps within the ventral lateral pulvinar whereas connectivity maps of motion-sensitive region TO overlapped with a medial portion of the ventral pulvinar. This spatial precision was clear in the group average while the distinction between these maps was less clear in individual subjects.

**HCP cortical connectivity profile**. As a control analysis to ensure that the connectivity patterns for the 39 areas used in the main analysis were not dependent on the particular ROIs, areal connectivity profiles were calculated using an 180 ROI parcellation of the whole cortical surface[32].

**Overlap analysis**. To compare the spatial localization of correlation patterns within the pulvinar, the overlap of pulvino-cortical connectivity maps was computed for all areas pairs with Dice's coefficient, $2|A \cap B| / (|A| + |B|)$. The group average map for each cortical area was threshold at $p < 0.05$ and binarized. Degree of overlap was assessed relative to the entire anatomical volume of the pulvinar.

**Multidimensional scaling**. To assess the structure of the pulvino-cortical connectivity, correlations were calculated between the pulvino-cortical connectivity maps of each cortical area. This high-dimension $39 \times 39$ connectivity matrix was converted to a dissimilarity (Euclidean distance) matrix and classical multidimensional scaling (MDS) was applied. The first two principal dimensions were visualized (Fig. 2). To compare the structure of the first two principal dimensions between hemispheres and across subjects, procrustes analysis was performed. For analysis of individual hemispheres, the left hemisphere was transformed to the right hemisphere space. Individual subjects were transformed to the group average space. Permutation tests were performed for the individual subject transformations. For each subject, the area labels for the two principal dimensions were scrambled and procrustes analysis was performed on the permuted data. This permutation was computed for each subject 1000 times. To test whether the group data were representative of individual subjects, two versions of the permutation test were performed. (1) Labels were permuted across all areas to test whether the structure of the individual subjects was similar to the pseudo-group structure (minus that subject) above chance. (2) Labels were permuted within each cluster from the group analysis to test whether the structure of individual subjects within each cluster was similar to the pseudo-group (minus that subject). The goodness of fit (sum of squared error; SSE) to the pseudo-group data for each subject's real data was compared with the goodness of fit for the permuted data.

**Clustering**. Clustering on the group average data was performed using a spectral (eigendecomposition) algorithm[28] from the Brain Connectivity Toolbox[29]. This algorithm automatically subdivides a (weighted) network into non-overlapping groups that maximizes the number of within-group links and minimizes the number of between group links. The algorithm starts with one group and continues to split the data into subgroups until additional splits yield no further minimization of between-group edges (as assessed by whether the signs of the eigenvectors are uniform). That is, the algorithm determines the optimal cluster size, which is an advantage over other clustering approaches, such as k-means. In practice, we found that k-means with a $k$ matched to the cluster size returned from this spectral clustering approach yielded qualitatively similar divisions that would not change interpretation of our results.

**Cortical distance**. Euclidean distances between the group average centroids for all 39 areas were calculated using AFNI's SurfDist. Cortical distance measures were correlated with Dice's coefficient (Sorensen index) and with the Euclidean distances between peaks in the pulvino-cortical connectivity maps.

**Experimental design and statistical analysis**. Thirteen subjects participated in resting state scans. Individual and group-level analyses were performed. Patterns of co-fluctuations in the BOLD signal were assessed within subject using Pearson correlation. Group-level statistical significance was assessed for the spatial pattern of pulvinar correlations by Fisher transforming individual subject correlations and performing one-sample, two-tailed t-tests. Data were corrected for multiple comparisons (FDR) at $p < 0.05$. MDS and clustering analyses were performed on the individual and group-average data. Pearson correlations were performed between the spatial pattern of correlations and cortical distance. Eleven subjects participated in a movie viewing experiment. Patterns of co-fluctuations in the BOLD signal were assessed between subjects using a Pearson's correlation. Five subjects participated in the contralateral tuning and attentional modulation experiment.

Response magnitudes and significance were estimated for each subject using a regression analysis. Group-level statistical significance was assessed across subjects using a one-sample, two-tailed t-tests. The data were corrected for multiple comparisons (FDR) at $p < 0.05$. A $d$ prime index was used to calculate degree of contralateral tuning and attentional modulation across subjects. Statistical significance was assessed across subjects using a two-sample, two-tailed $t$ tests. Sixteen subjects participated in the object localizer experiment. Voxel-wise response magnitudes and significance were estimated for each subject using a regression analysis. Voxel-wise group-level statistical significance was assessed across subjects using a mixed effects meta-analysis that models both within- and across- subject variability. For the ROI analysis, statistical significance was assessed across subjects using non-parametric testing (Wilcoxon signed rank test) and effect sizes (Cohen's d) were calculated.

## Data Availability

The data that support the findings of this study are available from the corresponding author upon reasonable request.

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

## Acknowledgements

This work was supported by grants from the National Institutes of Health (RO1 MH064043, RO1EY-017699, and T90DA-022763) and the National Science Foundation (BCS-1328270). We thank Christopher Honey, Stephanie McMains, and Ryan Mruczek for help with data collection.

## Author contributions

M.J.A., M.A.P., and J.C. collected data; M.J.A. and J.C. analyzed the data; M.J.A., M.A.P., J.C., and S.K. wrote the paper.

## Additional information

**Competing interests:** The authors declare no competing interests.

