## [Peer Review File · Nature Communications]

Reviewers' comments:

Reviewer #1 (Remarks to the Author):

The authors present a powerful study full of new results. They are the leading investigators in regard to the human pulvinar, and they have extensively reviewed the relevant primate studies here and elsewhere. They are expert on the methods used in this study, and it reflects high standards for publication. Thus, I feel that it is highly suitable for publication with few or no changes. That said, I have one concern. As it is, it is difficult for me to be certain how all of the results relate to proposed nuclei of the monkey pulvinar. It would be helpful to illustrate proposed nuclei in monkeys in relation to those in humans. As it is, I have the impression that the human ventral pulvinar with 2 retinotopic maps corresponds to the ventrolateral nucleus of the lateral pulvinar, and the central lateral nucleus of the inferior pulvinar, but I'm not sure if there is a functional difference between these two parts of the human ventral pulvinar. As for the dorsal pulvinar of humans, does this correspond to the proposed dorsomedial nucleus of the lateral pulvinar, the medial pulvinar, or both? As for the parts of the inferior pulvinar related to the MT complex, these nuclei may occupy a small caudal part of the human pulvinar? Do any of the present results reflect this part of the pulvinar? And do the present results suggest ways that the human pulvinar might differ from the monkey pulvinar? These are hard questions that may be difficult to address without more evidence. If so, what more is needed from monkey or human studies?

Reviewer #3 (Remarks to the Author):

Review of Arcaro et al.: Organizing principles of pulvino-cortical functional couplings in humans. This paper describes functional response properties of the human pulvinar, and how this structure interacts with the cortex. By and large the results are confirmatory as they corroborate previous evidence from the macaque, and previous human studies (including data collected by the same group). Specifically, Arcaro and colleagues observed a clear subdivision between dorsal and ventral parts of the pulvinar, whereby the dorsal pulvinar shows sensitivity to the temporal order of natural stimuli and is modulated by spatial attention whereas some degree of category-selectivity is observed in the ventral pulvinar. The functional differences were paralleled by differences in functional coupling with the cortex. This coupling, as indexed by correlations in fMRI signals between pulvinar and cortex, also differed between dorsal and ventral pulvinar. Arcaro et al. found stronger coupling between the dorsal pulvinar and more dorsal cortical regions, and ventral pulvinar with occipito-temporal cortex, mimicking the division between the dorsal and ventral visual streams.

I reviewed a previous version of this manuscript and the authors addressed most of my major concerns. The paper is well-written and the analyses are sound. The message is interesting and should appeal to a reasonably large audience -certainly since the putamen is gaining renewed scientific interest during the past couple of years. The only major leftover concern I have relates to the category selectivity in the putamen. In fact, this reviewer is not entirely convinced about a high degree of category selectivity. Voxels were found that responded more to faces than scenes (and more to faces than objects). Bodies were also tested, but apparently, they did not evoke differential activity patterns -there are no data shown unless I missed it. Together with the very small number of so-called 'selective' voxels in the putamen, this calls into question the degree of selectivity. It is certainly much smaller than that observed in ventral visual cortex to which the findings were related. I would like to see the activity profiles for all the stimuli tested and a selectivity index -and both should be compared with those of category selective regions in visual cortex. Also, faces and bodies are typically more 'interesting' stimuli than scenes. To what extent may the category-selectivity be explained by an arousal or attention account? I suggest that the authors put the degree of category-selectivity in perspective.

Minor:

1. How is the optimal number of clusters determined in the cluster analysis.

Reviewer #1 (Remarks to the Author):

As it is, it is difficult for me to be certain how all of the results relate to proposed nuclei of the monkey pulvinar. It would be helpful to illustrate proposed nuclei in monkeys in relation to those in humans.

We agree that such a visual comparison would be very helpful for relating the organization of the pulvinar across primate species. However, this would entail collecting corresponding datasets in the two species using identical methods. And such data sets do not currently exist in any lab to our knowledge. A different approach would be to have detailed anatomical data in humans to link them to our functional data. However, such detailed anatomical data are not available to identify individual pulvinar nuclei for direct comparisons with our functional data. Thus, we can only speculate on such correspondences from the locations of the foci of cortical correlations within the pulvinar (and their locations relative to each other). We hope that our specific responses below, and the corresponding additions to the manuscript text, help clarify how our imaging results relate in our interpretation to the organization in the monkey.

The BigBrain human histology atlas (Amunts et al. 2013, Science) provides a high-resolution (200 um) anatomical dataset of the pulvinar to compare with our functional activations. Though these data are from different individuals, this comparison provides some insight into the correspondence between subregions of the pulvinar and its functional organization. V1 correlations were situated within the ventral lateral extent of the pulvinar. Correlations with motion-sensitive area TO1 were situated within ventral medial sections of the pulvinar. Correlations with face-selective FFA were also situated in medial ventral portions of the pulvinar, but posterior to the peak TO1 correlations. Finally, correlations with parietal area IPS2 were focused within the dorsal pulvinar. The location of correlations within these histological sections is consistent with our own anatomical imaging (discussed below).

As it is, I have the impression that the human ventral pulvinar with 2 retinotopic maps corresponds to the ventrolateral nucleus of the lateral pulvinar, and the central lateral nucleus of the inferior pulvinar, but I'm not sure if there is a functional difference between these two parts of the human ventral pulvinar.

The reviewer is correct that the 2 prominent retinotopic maps in the human ventral pulvinar correspond to the maps in the ventrolateral nucleus of the lateral pulvinar and the central lateral nucleus of the inferior pulvinar. Similar to other primates, functional differences between these maps remain to be resolved. To our knowledge, there are no systematic studies investigating commonalities and/or differences among neuronal populations in these subdivisions. We found no differences in contralateral tuning, attentional modulation, or responses to movie stimuli between these two regions. Because this is a negative result that could be due to several factors (e.g., small volume / effective resolution), we hesitate to make any strong claims. We think it is likely that some functional differences should be present between these two maps, but higher spatial resolution MR protocols likely are needed to disentangle any functional differences. We have made the following clarification on page 14:

The ventral pulvinar showed strong tuning for contralateral visual space, consistent with previous work demonstrating that the human pulvinar contains at least two retinotopic maps^{32, 41, 59} and similar to the organization of the ventrolateral of the lateral pulvinar and central lateral nucleus in other primate species^{60, 61, 62, 63}.

As for the dorsal pulvinar of humans, does this correspond to the proposed dorsomedial nucleus of the lateral pulvinar, the medial pulvinar, or both?

Great question! We think both. We found that the correlations with the IPS areas are focused within lateral parts of the dorsal pulvinar. However, they are offset medially from the lateral edge of the pulvinar, suggesting that these foci are not solely contained within the lateral pulvinar. Though we do not have the resolution to anatomically distinguish individual pulvinar nuclei, it appears that the correlations with IPS areas straddle the dorsal lateral extent of the lateral pulvinar and the lateral border of the medial pulvinar. The spatial location of these correlation patterns (as seen in our Figure 5a) appears similar to the foci of anatomical connectivity between posterior IPS and the pulvinar in macaques as illustrated in this summary figure by Schmahmann & Pandya (2008, Cortex).

For inferior parietal (IPL) and the posterior cingulate / precuneus, the foci of pulvinar correlations were found medial to the IPS and FEF correlations and straddled the medial edge of the pulvinar. Thus, these correlations appear to be entirely within the medial pulvinar. While the correspondence of these cortical regions to the monkey is less clear than IPS, the cingulate in macaques is also known to project to the most medial parts of the pulvinar (Baleydier & Mauguier 1985 J Comp Neurol).

We have clarified this relationship on pages 9-10:

The foci of these correlations were situated just medial to the lateral edge of the dorsal pulvinar. In contrast, the pulvino-cortical connectivity maps of other dorsal areas appeared to be spatially distinct. For example, the foci of inferior parietal lobule (IPL) and posterior cingulate / precuneus (PreC) correlations fell medial to the IPS and FEF foci and straddled the medial edge of the pulvinar.

And on page 15:

Functional coupling between the pulvinar and dorsal attention network areas (IPS1-4, FEF, IFS) appeared to be situated within lateral parts of the dorsal pulvinar and potentially corresponds to the border between the dorsal lateral nucleus and the lateral extent of the medial pulvinar, similar to other primates³⁰. In contrast, the foci of functionally coupling with the cingulate and IPL were situated entirely within the medial pulvinar, similar to other primates⁵⁰.

As for the parts of the inferior pulvinar related to the MT complex, these nuclei may occupy a small caudal part of the human pulvinar? Do any of the present results reflect this part of the pulvinar?

We found that the foci of correlations with cortical areas TO-1 and TO-2 (likely homologues to the MT complex in monkeys) are located within the ventral pulvinar and medial to the foci of V1-V3 correlations (Figure 6) as well as medial to the two retinotopic maps we previously reported. This is consistent with the notion that the human MT complex is linked with medial parts of the inferior pulvinar. However, the relationship between Plm and human MT would best be resolved if we could anatomically differentiate subdivisions of the inferior pulvinar and relate to the functional organization. We discuss the relation of human TO-pulvinar coupling to monkey MT-pulvinar connectivity in the results on page 14.

Interestingly, the foci of correlations with neighboring cortical areas such as body-selective EBA and face-selective pSTS were found within the ventral medial pulvinar proximal to the TO correlations. Though homologues of these regions in the monkey remain to be resolved, the cortical regions surrounding the MT complex in monkeys (e.g., FST and STP) also project to parts of the pulvinar proximal to the medial nucleus of the inferior pulvinar. Together, these data suggest that the specific connectivity between MT and the medial nucleus of the inferior pulvinar is situated within a broader cortical topography (e.g., Fig. 6 in Shipp 2003, Phil. Trans. R. Soc. Lond. B). We now discuss this on page 14:

The functional coupling between the pulvinar and TO1/2 appears to be situated within a broader topography linking lateral temporal regions and the ventral medial pulvinar. The foci of functional coupling for lateral temporal cortical regions, EBA and pSTS, were also found within medial portions of the ventral pulvinar, proximal to the TO foci. Though monkey homologues of these regions remain to be resolved, cortical regions surrounding MT in monkeys (e.g., FST and STP) are interconnected with parts of the pulvinar proximal to MT connections (i.e., adjacent to Plm)²⁶. These data suggest that discrete pulvinocortical connections (such as between MT and Plm^{27, 28, 29}) are embedded within a larger framework that preserves cortical topography⁶.

And do the present result suggest ways that the human pulvinar might differ from the monkey pulvinar?

Great question! While differences likely exist between species, we know that the general anatomical organization of pulvinar nuclei are similar between humans and monkeys (e.g., Cola et al. 1999, Neuroreport). As such, we do not expect dramatic differences in the overall organization. Rather, our approach is to ask how the pulvinar can incorporate human-specific functions while preserving this global structure. For example, humans have regions in anterior parietal and lateral temporal cortex involved in the processing of tools (referred to as antPPC and latTemp in our study). We find that these regions are most strongly coupled with the dorsal pulvinar. The foci of these correlations were located near the foci of correlations for other areas proximal to anterior parietal cortex, but these two areas were grouped into a separate cluster, suggesting that the dorsal pulvinar has expanded to allow for interactions with human-specific networks (e.g., tool processing) while also preserving the overall topographic layout of cortical connectivity. Our results shed light on how the human pulvinar is capable of interacting with these additional areas while preserving the global structure that is consistent across primate species such as chimps and old world monkeys. We now include a discussion on page15:

The pulvinar incorporates human-specific adaptations. The dorsal medial pulvinar was functionally coupled with anterior parietal (antPPC) and lateral temporal (latTemp) cortical regions involved in the processing of tools. The foci of these correlations were located near the foci of other areas proximal to anterior parietal cortex, but these two areas (along with IPS5) were grouped into a separate cluster, suggesting that the dorsal pulvinar has expanded to allow for interactions with human-specific cortical networks. Despite a substantial increase in size across primates⁸², the human pulvinar maintains an organization of nuclei similar to other primate species⁸². Together, these data suggest that the expansion of individual nuclei allows for species-specific functional adaptations within the pulvinar while preserving its global organization across primates.

In addition to identifying functional coupling with human-specific cortical areas, we found that the human pulvinar was functionally diverse and was sensitive to object-category visual information and the temporal structure of visual input. While recent studies in macaques have shown sensitivity to complex shapes (including faces) in the pulvinar (Nguyen et al. 2016 Front. Neurosci.), we show that this information is spatially localized. Such functional clustering is a ubiquitous organizing principle of cortex. Here, we show that the same principle exists in the thalamus despite its lack

of a cortical surface and columnar organization. Further, cortical projections heavily overlap in the pulvinar (e.g., Shipp 2001), so it is notable that this overlap is not so great as to blur out input from functionally-clustered regions in IT (e.g., face clusters). We have expanded our discussion of this on page 15:

This is particularly notable given the extensive overlap of cortical projections throughout the pulvinar^{6, 20, 56}, which could have had the effect of blurring out such functional specificity. While emphasis is often placed on the pulvinar's role in visual attention, our results illustrate a distinct role in object vision, and more broadly illustrate that ubiquitous principles of cortical organization (e.g., functional clustering) hold for subcortex despite considerable differences in architecture (e.g., lack of columnar organization).

Lastly, one intriguing result is that the pulvinar coupling with human motion-sensitive areas TO1/2 was more strongly associated with the coupling of other the ventral stream than dorsal stream areas. This is in apparent contrast to the monkey, where MT is more strongly associated with the dorsal stream based on patterns of anatomical cortico-cortical connectivity (Young 1992, Nature) and cortico-cortical functional coupling (Arcaro et al. 2018, eLife). Consistent with our current findings, a recent human neuroimaging study found that TO was more strongly associated with other ventral stream areas based on cortico-cortical activity patterns (Haak & Beckmann 2018, Cortex). Since comparable analyses have not been performed on pulvino-cortical functional coupling in monkeys, it remains unclear whether these data reflect differences between species or methodologies. Future work will be needed to resolve the relationship of pulvino-cortical connectivity between species. It's worth noting that our clustering did not solely separate ventral and dorsal cortex based on cortical distance as the lateral temporal tool area was more strongly associated with the dorsal pulvinar. As mentioned in our response to an earlier point raised by the reviewer, we discuss the relation of human TO-pulvinar coupling to monkey MT-pulvinar connectivity in the results on page 14.

These are hard questions that may be difficult to address without more evidence. If so, what more is needed from monkey or human studies?

Higher MR imaging resolution would certainly help. Perhaps more essential would be to have equally precise anatomical imaging techniques to compare with the variety of functional responses we find across the pulvinar. Having such structure-function correspondences would also provide better ground for resolving similarities / differences across primate species. We add a note about this future work on page 16:

Lastly, the relationship between the functional organization and individual nuclei within the human pulvinar remains to be resolved. Though the localization of functional coupling for individual cortical areas within the pulvinar provides insight into such structure-function relationships, development of high-resolution anatomical image techniques that allow for the precise delineation of nuclei is needed and would facilitate further comparisons across primate species.

Reviewer #3 (Remarks to the Author):

The only major leftover concern I have relates to the category selectivity in the putamen. In fact, this reviewer is not entirely convinced about a high degree of category selectivity. Voxels were found that responded more to faces than scenes (and more to faces than objects). Bodies were also tested, but apparently, they did not evoke differential activity patterns -there are no data shown unless I missed it. Together with the very small number of so-called 'selective' voxels in the putamen, this calls into question the degree of selectivity. It is certainly much smaller than that observed in ventral visual cortex to which the findings were related.

The reviewer's point regarding the degree of selectivity is well taken. We agree with the reviewer that additional data was needed to support the claim of category selectivity within the pulvinar. To address this concern, we performed new analyses, include new figures, and made extensive changes to the manuscript text. To evaluate the degree of selectivity within this posterior medial region of the pulvinar, we extracted the beta coefficients for faces, bodies, objects, scrambled images, and scenes. To avoid circularity issues, we used a leave-one-out approach, where we identified an ROI for faces > scenes within the posterior pulvinar based on n-1 subjects then extracted the mean beta coefficients from the left our subject. We did this for each subject. In Figure 1c, we now plot the mean, standard error, and individual subject betas for faces and scenes. Dashed lines indicate the correspondence of face and scene betas for each subject. As now described on pages 4, we found a significantly larger response to the faces than scenes in this posterior region of the pulvinar ($p = 0.011$, two-tailed):

Face selectivity was apparent at the individual subject level. The mean beta coefficients were extracted from this region in individual subjects using a leave-one-out analysis (Methods: Category localizer ROI analysis) to avoid issues of circularity. Faces consistently evoked a larger response than scenes in individual subjects (Fig. 1c, $p < 0.05$).

In Supplementary Figure 1a, we now show the mean betas, standard error, and individual subject betas for all five stimulus categories in the posterior pulvinar. In addition to the face-minus-scene coefficient contrast, we found significantly greater BOLD responses to faces than scrambled images and as well as to bodies than objects, scrambled images, and scenes ($p < 0.05$). The greater face response compared to objects was marginal ($p = 0.06$).

We did not find a significant difference between responses to faces and bodies, though we would not expect to see such a difference given our imaging resolution, the small size of the pulvinar, and the close proximity of face- and body- selective cortical regions (FFA and FBA). As stated above, this posterior medial region of the pulvinar was functionally coupled with the FFA. The FFA is located adjacent to, and partially overlapping with, FBA. These two cortical regions appear to partially overlap even at high MR imaging resolutions (e.g., 1.4x1.4x2mm; Schwarzlose et al. 2005, JNeuro). As such, we would not expect to differentiate between face- and body- selective regions at our scanning resolution. A much higher spatial resolution, likely at higher field strength, would be needed to resolve any spatial differences in their localization.

We have made the following edits to page 5:

Across all five categories, faces and bodies evoked the largest responses in this region as compared with scenes, scrambled images, and objects (Supplementary Fig. 1a). The lack of a significant differential faces and bodies may reflect the presence of partially overlapping face- and body-selective regions within the posterior pulvinar that are at a scale finer than our imaging resolution. Indeed, face- (FFA) and body- (FBA) selective regions within fusiform cortex overlap even at high spatial imaging resolutions⁴⁴.

Further, we found that this posterior medial region of the pulvinar was coupled with face-selective cortical regions during rest (Figure 6). To probe this link with resting state correlations further, we performed a new analysis that compared the spatial pattern of coefficients from a face-minus-scene coefficient contrast within the pulvinar with the resting-state correlation patterns between the pulvinar and each of the 39 cortical areas explored in our study. Consistent with the category selectivity mapping results, we found a significant correlation between the face-minus-scene coefficient contrast and the pulvinar correlations for face-selective regions FFA, AT, and pSTS ($r_s = 0.35, 0.35,$ and 0.46 , respectively). Further, the correlation patterns for these areas were more similar to the face-minus-scene coefficients than any of the other 36 cortical areas tested (Supplementary Figure 2). We have edited the results on page 12 accordingly:

Notably, the FFA's functional coupling was localized to the same postero-medial region of the ventral pulvinar that showed face-selective activity (Fig. 1d) and the spatial patterns of functional coupling for face-selective areas FFA, AT, and pSTS were each correlated with the spatial pattern of the face-minus-scene coefficient contrasts ($r_s > 0.34$) more than any of the other 36 areas (Supplementary Fig. 2), demonstrating that pulvino-cortical coupling during rest is predictive of pulvinar activity during visual stimulation.

Altogether, these results indicate that a spatially focal region of the medial posterior pulvinar is involved in the processing of faces and bodies visual images.

I would like to see the activity profiles for all the stimuli tested and a selectivity index -and both should be compared with those of category selective regions in visual cortex.

To facilitate comparisons between the pulvinar face-responses and what is typically observed in cortex, we now provide cortical surface maps for face-minus-scene coefficient contrast maps (Supplementary Figure 1b). We use the same threshold as was used in the pulvinar analysis ($p < 0.05$ FDR-corrected). Several visual category-selective cortical regions are apparent, including face-selective regions OFA, FFA(1/2), AT, pSTS, and the amygdala (which can be partially seen on the medial surface reconstructions), as well as scene-selective regions PPA, TOS, RSC. In Figure 1c, we show face and scene beta coefficients with interconnecting lines for each subject.

However, we do not think a direct comparison of response magnitudes between cortex and the pulvinar is appropriate. Given the large difference in our effective imaging resolution (due to the much smaller size of the pulvinar), such comparisons could lead to false inferences. For example, imagine we scanned ventral temporal cortex at a resolution that only yielded 10-15 voxels covering the FFA. Many of those voxels would partial volume neighboring non-face-selective cortex. The resulting selectivity indices would be much smaller than at conventional imaging resolutions (1.5 – 3mm) due to the extensive partial voluming and any inferences about the degree of selectivity would be invalid. As such, we are not making a claim about the degree of face-selectivity within the human

pulvinar, but simply that face-selectivity is present and it is spatially clustered. The later point is particularly important. The presence of this type of functional clustering within the pulvinar is entirely novel and has important implications for the functional organization of the pulvinar just as it has had for the organization of cortex. Functional clustering along the cortical surface (across cortical columns) is a well-established principle of cortical organization. The architecture of the pulvinar (and subcortex in general) is vastly different (e.g., no columnar organization), yet we show this basic principle holds. As noted in our response to reviewer #1, we expanded our discussion of this point on page 15:

This is particularly notable given the extensive overlap of cortical projections throughout the pulvinar^{6, 20, 56}, which could have had the effect of blurring out such functional specificity. While emphasis is often placed on the pulvinar's role in visual attention, our results illustrate a distinct role in object vision, and more broadly illustrate that ubiquitous principles of cortical organization (e.g., functional clustering) hold for subcortex despite considerable differences in architecture (e.g., lack of columnar organization).

Also, faces and bodies are typically more 'interesting' stimuli than scenes. To what extent may the category-selectivity be explained by an arousal or attention account?

This is certainly an important point. Faces are typically thought of as more 'interesting' stimuli. However, we do not think that arousal or attention signals can account for the face-selective activity in the pulvinar. First, face-selectivity was not observed in the dorsal pulvinar, which was functionally coupled with the dorsal attention network (IPS1-4, FEF, IFS). If greater attention was paid to faces or bodies than to other stimulus categories (or scrambled images), then this should have been reflected in areas associated with the attention network. Second, this region of the medial posterior pulvinar was functionally coupled with ventral temporal cortex and the resting state patterns of the face-selective cortical regions were most correlated with the spatial pattern of face-minus-scene contrast. Third, we do not find significant face-selectivity within the dorsal attention network (Supplementary Figure 1b), which would be expected if this was an arousal or attention signal. We now address this point on page 5:

Though faces and bodies can be thought of as more 'interesting' or attention-grabbing than some of the other stimuli tested, it is unlikely that the effects observed in the pulvinar are driven by arousal or attentive signals. If greater attention had been paid to faces or bodies than to other stimulus categories (or scrambled images) then this should have been reflected in areas associated with the attention network (IPS1-4, FEF, IFS).

Minor:

1. How is the optimal number of clusters determined in the cluster analysis.

Clustering was performed using the `modularity_und.m` function from the Brain Connectivity Toolbox, which is an MATLAB implementation of Newman's modularity algorithm. (Newman 2006, PNAS). This algorithm finds an optimal community structure that maximizes the number of within-group edges, and minimizes the number of between-group edges through spectral decomposition. The algorithm starts with one group and continues to split the data into subgroups until the additional splits yield no further minimization of between-group edges (assessed by whether the signs of the eigenvectors are uniform). The algorithm returns a Q coefficient which is a statistic that quantifies the degree to which the network can be clearly subdivided into groups. We now clarify this on page 22.

Reviewers' comments:

Reviewer #1 (Remarks to the Author):

The authors have addressed my comments. I have no further concerns.

Reviewer #3 (Remarks to the Author):

The authors partially addressed my concerns.

I'm still a bit puzzled by the category selectivity. I think if a non-parametric test would be performed on the data of Suppl. Fig 1, differences between categories will probably not be significant. Please test this formally. Such an analysis is probably warranted as, at first sight at least, the data don't seem to be normally distributed.

At the very least the weakness of the category selectivity (if it survives rigorous testing) should be acknowledged. I only follow partially the resolution argument of the authors: in case this cannot be resolved with the current technological limitations, one shouldn't make too big claims....

In addition to the reviewer's requests, we added subject numbers in the Methods section to clarify the relationship between subjects and experiments (p. 17). Also, we previously reported that 15 subjects participated in the object localizer experiment. In fact, 16 subjects participated in the experiment. This does not affect any of the results. All analyses and statistical tests were performed correctly using all 16 subjects. We have corrected the subject description in the manuscript and double checked that the subject counts for all other experiments are correctly listed.

Reviewer #3 (Remarks to the Author):

I'm still a bit puzzled by the category selectivity. I think if a non-parametric test would be performed on the data of Suppl. Fig 1, differences between categories will probably not be significant. Please test this formally. Such an analysis is probably warranted as, at first sight at least, the data don't seem to be normally distributed. At the very least the weakness of the category selectivity (if it survives rigorous testing) should be acknowledged. I only follow partially the resolution argument of the authors: in case this cannot be resolved with the current technological limitations, one shouldn't make too big claims....

As the reviewer requested, we tested for normality in our object localizer data. The distribution of responses to faces, bodies, scrambled images, and scenes did not significantly differ from a normal distribution (all p s > 0.3061). However, the distribution of responses to the object category were not normally distributed ($p = 0.0019$). We took the reviewer's advice and performed nonparametric testing (Wilcoxon signed rank test). We found that all significant effects held. Face vs scenes: $z = 3.1025$, $p = 0.0019$. Faces vs. scrambled images: $z = 2.7406$, $p = 0.0061$. Bodies vs scenes: $z = 2.6371$, $p = 0.0084$. Bodies vs. objects: $z = 2.6371$, $p = 0.0084$. Bodies vs. scram: $z = 2.4303$, $p = 0.0151$. The p value for faces vs. objects (previously $t(15)=2.0331$, $p = 0.0601$) was slightly larger: $z = 1.7581$, $p = 0.0787$, but remains trending. We have updated the manuscript and figures accordingly. In addition, we calculated the effect size for the face vs. scene contrast: Cohen's $d = 0.63$, which indicates a medium-sized effect.

In the previous revision, we stated that the effects in the pulvinar appear weaker than cortical effects (p. 5), but we think caution should be exercised in making direct comparisons. We outlined several issues with such comparisons (p. 5). As such, we have been very careful not to make any qualitative statements regarding the degree of these effects. Our conclusion that the posterior medial pulvinar is involved in high-level visual form processing is supported by the following three converging results: (1) significantly greater responses to faces vs. scenes (and scrambled images) within the posterior medial pulvinar, (2) functional coupling between this region and face-selective cortical regions at rest, and (3) functional coupling patterns between face-selective regions (FFA, AT, and pSTS) and the pulvinar that predict, more than any other cortical region tested, the face vs. scene beta coefficient patterns.

REVIEWERS' COMMENTS:

Reviewer #3 (Remarks to the Author):

I thank the authors for performing the additional analysis. I have no further comments and wish to congratulate them with this nice piece of work.